# Ensemble Kalman Diffusion Guidance: A Derivative-free Method for Inverse Problems

**Hongkai Zheng**                                              *hzzheng@caltech.edu*
*Caltech*

**Wenda Chu**                                                    *wchu@caltech.edu*
*Caltech*

**Austin Wang**                                                *akwang@caltech.edu*
*Caltech*

**Nikola B. Kovachki**                                      *nkovachki@nvidia.com*
*NVIDIA*

**Ricardo Baptista**                                              *rsb@caltech.edu*
*Caltech*

**Yisong Yue**                                                    *yyue@caltech.edu*
*Caltech*

**Reviewed on OpenReview:** *https://openreview.net/forum?id=XPEEsKneKs*

## Abstract

When solving inverse problems, one increasingly popular approach is to use pre-trained diffusion models as plug-and-play priors. This framework can accommodate different forward models without re-training while preserving the generative capability of diffusion models. Despite their success in many imaging inverse problems, most existing methods rely on privileged information such as derivative, pseudo-inverse, or full knowledge about the forward model. This reliance poses a substantial limitation that restricts their use in a wide range of problems where such information is unavailable, such as in many scientific applications. We propose Ensemble Kalman Diffusion Guidance (EnKG), a derivative-free approach that can solve inverse problems by only accessing forward model evaluations and a pre-trained diffusion model prior. We study the empirical effectiveness of EnKG across various inverse problems, including scientific settings such as inferring fluid flows and astronomical objects, which are highly non-linear inverse problems that often only permit black-box access to the forward model. We open-source our code at https://github.com/devzhk/enkg-pytorch.

## 1 Introduction

The idea of using pre-trained diffusion models (Song et al., 2020; Ho et al., 2020) as plug-and-play priors for solving inverse problems has been increasingly popular and successful across various applications including medical imaging (Song et al., 2021; Sun et al., 2023), image restoration (Chung et al., 2022b; Wang et al., 2022), and image and music generation (Rout et al., 2024b; Huang et al., 2024). A key advantage of this approach is its flexibility to accommodate different problems without re-training while maintaining the expressive power of diffusion models to capture complex and high-dimensional prior data distributions. However, most existing algorithms rely on privileged information of the forward model, such as its derivative (Chung et al., 2022a; Song et al., 2023b), pseudo-inverse (Song et al., 2023a), or knowledge of its parameterization (Chung et al., 2023a). This reliance poses a substantial limitation that prevents their

use in problems where such information is generally unavailable. For instance, in many scientific applications (Oliver et al., 2008; Evensen & Van Leeuwen, 1996; Iglesias, 2015), the forward model consists of a system of partial differential equations whose derivative or pseudo-inverse is generally unavailable or even undefined.

The goal of this work is to develop an effective method that only requires black-box access to the forward model and pre-trained diffusion model for solving general inverse problems. Such an approach could significantly extend the range of diffusion-based inverse problems studied in the current literature, unlocking a new class of applications – especially many scientific applications. The major challenge here arises from the difficulty of approximating the gradient of a general forward model with only black-box access. The standard zero-order gradient estimation methods are known to scale poorly with the problem dimension (Berahas et al., 2022).

To develop our approach, we first propose a generic prediction-correction (PC) framework using an optimization perspective that includes existing diffusion guidance-based methods (Chung et al., 2022a; Song et al., 2023b;a; Peng et al., 2024; Tang et al., 2024) as special cases. The key idea of this PC framework is to decompose diffusion guidance into two explicitly separate steps, unconditional generation (i.e., sampling from the diffusion model prior), and guidance imposed by the observations and forward model. This modular viewpoint allows us to both develop new insights of the existing methods, as well as to introduce new tools to develop a fully derivative-free guidance method. Our approach, called Ensemble Kalman Diffusion Guidance (EnKG), uses an ensemble of particles to estimate the guidance term while only using black-box queries to the forward model (i.e., no derivatives are needed), using a technique known as statistical linearization (Evensen, 2003; Schillings & Stuart, 2017) that we introduce to diffusion guidance via our PC framework.

**Contributions**

- We present a generic prediction-correction (PC) framework that provides an alternative interpretation of guided diffusion, as well as additional insights of the existing methods.

- Building upon the PC framework, we propose Ensemble Kalman Diffusion Guidance (EnKG), a fully derivative-free approach that leverages pre-trained model in a plug-and-play manner for solving general inverse problems. EnKG only requires black-box access to the forward model and can accommodate different forward models without any re-training.

- We evaluate on various inverse problems including the standard imaging tasks and scientific problems like the Navier-Stokes equation and black-hole imaging. On more challenging tasks, such as nonlinear phase retrieval in standard imaging and the scientific inverse tasks, our proposed EnKG outperforms baseline methods by a large margin. For problems with very expensive forward models (e.g., Navier-Stokes equation), EnKG also stands out as being much more computationally efficient than other derivative-free methods.

## 2 Background & Problem Setting

**Problem setting** Let $G : \mathbb{R}^n \to \mathbb{R}^m$ denote the forward model that maps the true unobserved source $\boldsymbol{x}$ to observations $\boldsymbol{y}$. We consider the following setting:

$$\boldsymbol{y} = G(\boldsymbol{x}) + \xi, \quad \boldsymbol{x} \in \mathbb{R}^n, \boldsymbol{y}, \xi \in \mathbb{R}^m \tag{1}$$

where we only have black-box access to $G$ (generally assumed to be non-linear). $\xi$ represents the observation noise which is often modeled as Gaussian, i.e., $\xi \sim \mathcal{N}(0, \Gamma)$, and $\boldsymbol{y}$ represents the observation. Solving the inverse problem amounts to inverting Eq. (1), i.e., finding the most likely $\boldsymbol{x}$ (MAP inference) or its posterior distribution $P(\boldsymbol{x}|\boldsymbol{y})$ (posterior inference) given $\boldsymbol{y}$. This inverse task is often expressed via Bayes's rule as $p(\boldsymbol{x}|\boldsymbol{y}) \propto p(\boldsymbol{y}|\boldsymbol{x}) \cdot p(\boldsymbol{x})$. Here $p(\boldsymbol{x})$ is the prior distribution over source signals (which we instantiate using a pre-trained diffusion model), and $p(\boldsymbol{y}|\boldsymbol{x})$ is defined as (1). Because we only have black-box access to $G$, we can only sample from $p(\boldsymbol{y}|\boldsymbol{x})$, and do not know its functional form. For simplicity, we focus on finding the MAP estimate: $\arg\max_{\boldsymbol{x}} p(\boldsymbol{y}|\boldsymbol{x}) \cdot p(\boldsymbol{x})$.

**Diffusion models**    Diffusion models (Song et al., 2020; Karras et al., 2022) capture the prior $p(\boldsymbol{x})$ implicitly using a diffusion process, which includes a forward process and backward process. The forward process transforms a data distribution $\boldsymbol{x}_0 \sim p_{\text{data}}$ into a Gaussian distribution $\boldsymbol{x}_T \sim \mathcal{N}(0, \sigma^2(T)\boldsymbol{I})$ defined by a pre-determined stochastic process. The Gaussian distribution is often referred to as noise, and so the forward process ($t$ going from 0 to $T$) is typically used to create training data (iteratively noisier versions of $\boldsymbol{x}_0 \sim p_{\text{data}}$) for the diffusion model. The backward process ($t$ going from $T$ to 0), which is typically learned in a diffusion model, is the standard generative model and operates by sequentially denoising the noisy data into clean data, which can be done by either a probability flow ODE or a reverse-time stochastic process. Without loss of generality, we consider the following probability flow ODE since every other probability flow ODE is equivalent to it up to a simple reparameterization as shown by Karras et al. (2022):

$$\mathrm{d}\boldsymbol{x}_t = -\dot{\sigma}(t)\sigma(t)\nabla_{\boldsymbol{x}_t} \log p_t(\boldsymbol{x}_t)\mathrm{d}t. \tag{2}$$

Training a diffusion model amounts to training the so-called score function $\nabla_{\boldsymbol{x}_t} \log p_t(\boldsymbol{x}_t)$, which we assume is already completed (and not the focus of this paper). Given a trained diffusion model, we can sample $p(\boldsymbol{x})$ by integrating (2) starting from random noise.

**Diffusion guidance for inverse problems**    As surveyed in Daras et al. (2024), arguably the most popular approach to solving inverse problems with a pre-trained diffusion model is guidance-based (Chung et al., 2022a; Wang et al., 2022; Kawar et al., 2022; Song et al., 2023a; Zhu et al., 2023; Rout et al., 2023; Chung et al., 2023b; Tang et al., 2024). Guidance-based methods are originally interpreted as the conditional reverse diffusion process targeting the posterior distribution. For ease of notation and clear presentation, we use the probability flow ODE to represent the reverse process and rewrite it with Bayes Theorem.

$$\begin{aligned}
\mathrm{d}\boldsymbol{x}_t &= -\dot{\sigma}(t)\sigma(t)\nabla_{\boldsymbol{x}_t} \log p_t(\boldsymbol{x}_t|\boldsymbol{y})\mathrm{d}t, \\
&= -\dot{\sigma}(t)\sigma(t)\nabla_{\boldsymbol{x}_t} \log p_t(\boldsymbol{x}_t)\mathrm{d}t - \dot{\sigma}(t)\sigma(t)\nabla_{\boldsymbol{x}_t} \log p_t(\boldsymbol{y}|\boldsymbol{x}_t)\mathrm{d}t,
\end{aligned} \tag{3}$$

where $\nabla_{\boldsymbol{x}_t} \log p_t(\boldsymbol{x}_t)$ is the unconditional score and the $\nabla_{\boldsymbol{x}_t} \log p_t(\boldsymbol{y}|\boldsymbol{x}_t)$ is the guidance from likelihood. In practice, the unconditional score is approximated by a pre-trained diffusion model $s_\theta(\boldsymbol{x}_t, t)$. The likelihood term, $p_t(\boldsymbol{y}|\boldsymbol{x}_t) = \int_{\boldsymbol{x}_0} p(\boldsymbol{x}_t|\boldsymbol{x}_0)p(\boldsymbol{y}|\boldsymbol{x}_0)\mathrm{d}\boldsymbol{x}_0$, is computationally intractable as it requires integration over all possible $\boldsymbol{x}_0$. Various tractable approximations have been proposed in the literature, which we denote as $\hat{p}_t(\boldsymbol{y}|\boldsymbol{x}_t)$. The corresponding approximated reverse process is:

$$\mathrm{d}\boldsymbol{x}_t = -\dot{\sigma}(t)\sigma(t)s_\theta(\boldsymbol{x}_t, t)\mathrm{d}t - w_t\nabla_{\boldsymbol{x}_t} \log \hat{p}_t(\boldsymbol{y}|\boldsymbol{x}_t)\mathrm{d}t, \tag{4}$$

where $w_t$ is the adaptive time-dependent weight. The design of $w_t$ in Eq. (4) varies across different methods but it is typically not related to $\dot{\sigma}(t)\sigma(t)$ that Eq. (3) suggests, which makes it hard to interpret from a posterior sampling perspective. In this paper, we will take an optimization perspective develop a useful interpretation for designing our proposed algorithm.

One key issue with Eq. (4) is that many algorithms for sampling along Eq. (4) assume access to the gradient $\nabla_{\boldsymbol{x}_t} \log \hat{p}_t(\boldsymbol{y}|\boldsymbol{x}_t)$. When this gradient is unavailable (e.g., only black-box access to $\hat{p}_t(\boldsymbol{y})$), then one must develop a derivative-free approach, which is our core technical contribution.

Two existing derivative-free diffusion guidance methods are stochastic control guidance (SCG) (Huang et al., 2024), and diffusion policy gradient (DPG) (Tang et al., 2024). Both SCG and DPG are developed from the stochastic control viewpoint, and guides the diffusion process via estimating a value function, which can be challenging to estimate well (as seen in our experiments).

## 3   Related work

**Ensemble Kalman methods**    Ensemble Kalman methodology was first introduced by Evensen (1994) in the context of data assimilation and later revisited by Iglesias et al. (2013) for inverse problems from an optimization perspective, resulting in the derivative-free algorithm known as Ensemble Kalman Inversion (EKI). Subsequent advancements include momentum-based EKI for training neural networks (Kovachki & Stuart, 2019) and various regularization techniques to improve stability and efficiency (Iglesias, 2016; Chada

et al., 2020). While the original derivations of these methods (Evensen, 1994; Iglesias et al., 2013) may not explicitly highlight it, the overarching idea behind the ensemble Kalman methodology can be interpreted as linearization of the forward model $G$ as elucidated in Schillings & Stuart (2017); Law et al. (2016). In essence, we approximate the potentially complex forward model $G(x)$ with the best-fitting linear surrogate model $y = Ax + b$. In other words, we seek the minimizers of the following optimization problem:

$$\min_{A,b} \mathbb{E}_x \| G(x) - Ax - b \|_2^2,$$

which has closed-form solutions given by

$$A = \mathbb{E}_x [(G(x) - \mathbb{E}_x G(x)) x^\top] C_{xx}^{-1}, b = \mathbb{E}_x [G(x)] - \mathbb{E}_x [Ax],$$

where $C_{xx}^{-1}$ is the pseudoinverse of the covariance matrix. Our approach builds upon this linear surrogate model to approximate the forward model in our likelihood step defined in Eq. (14), enabling a fully derivative-free algorithm. Furthermore, prior works (Bergou et al., 2019; Chada & Tong, 2022) establish the convergence results for certain EKI variants in the non-linear setting. However, these proofs do not directly apply to our algorithm. Instead, we develop tailored convergence results for our analysis.

**Derivative-free optimization** Traditional derivative-free optimization (DFO) algorithms include direct search, which includes the coordinate search, stochastic finite-difference approximations of the gradient, Nelder-Mead simplex methods, and model-based methods; see Berahas et al. (2022) for an overview. Among modern DFO techniques, Gaussian smoothing methods (Nesterov & Spokoiny, 2017) have demonstrated robust empirical performance (Salimans et al., 2017). These gradient estimates can be plugged into gradient-based algorithms directly, which we use to establish strong baselines in this paper.

**Predictor-Corrector method for diffusion models** The term "predictor-corrector" has been used in several prior works on diffusion model sampling, but their objectives and mechanisms differ from our PC framework. Song et al. (2020) introduces a Predictor-Corrector sampler for the sampling of diffusion models, where both the predictor and corrector aim to sample from the same target distribution by simulating different stochastic processes (reverse-time SDE and annealed Langevin dynamics, respectively). Lezama et al. (2022) then extends this framework to discrete space. More recently, Bradley & Nakkiran (2024) applies a similar perspective to classifier-free guidance, using PF-ODE solver (DDIM) as the predictor and Langevin dynamics as corrector, in order to sample from the gamma-powered data distribution. Other works, such as Zhao et al. (2023) and Zhao et al. (2024), draw inspiration from predictor-corrector methods in the literature of classical numerical ODE solvers, focusing on solving the probability flow ODE (PF-ODE) with higher-order accuracy and adaptive step sizes. These existing PC methods aim to sample from the diffusion model, which is related to the prediction step (sampling from diffusion prior) in our framework but irrelevant to our correction step that interacts with the forward model $G$.

## 4 Method

To develop our Ensemble Kalman Diffusion Guidance (EnKG) method, we first provide an interpretation of diffusion guidance through the lens of the prediction-correction framework. EnKG can be viewed as an instantiation which enables derivative-free approximation of the guidance term.

### 4.1 Prediction-correction interpretation of guidance-based methods

Inspired by the idea of the Predictor-Corrector continuation method in numerical analysis (Allgower & Georg, 2012), we show how to express the guidance-based methods within the following prediction-correction framework. Suppose the time discretization scheme is $T = t_0 > t_1 \cdots > t_N = 0$. Let $w_i = w_{t_i}$ for light notation. As illustrated in Algorithm 1, guidance-based methods for inverse problems can be summarized into the following steps.

---

**Algorithm 1** Generic Guidance-based Method (ODE version)

---

**Require:** $G, \boldsymbol{y}, s_\theta, \{t_i\}_{i=1}^N, \{w_i\}_{i=1}^N$

1: **sample** $\boldsymbol{x}_0 \sim \mathcal{N}(0, \sigma^2(t_0)\boldsymbol{I})$
2: **for** $i \in \{0, \dots, N-1\}$ **do**
3:     $\boldsymbol{x}_i' \leftarrow \boldsymbol{x}_i - \dot{\sigma}(t_i)\sigma(t_i)s_\theta\left(\boldsymbol{x}_i, t_i\right)\left(t_{i+1} - t_i\right)$      ▷ Prior prediction step
4:     $\log \hat{p}(\boldsymbol{y}|\boldsymbol{x}_t) \approx \log p(\boldsymbol{y}|\boldsymbol{x}_t)$      ▷ Log-likelihood estimation
5:     $\boldsymbol{x}_{i+1} \leftarrow \arg\min_{\boldsymbol{x}_{i+1}} \frac{\|\boldsymbol{x}_{i+1} - \boldsymbol{x}_i'\|_2^2}{2w_i} - \log \hat{p}(\boldsymbol{y}|\boldsymbol{x}_{i+1})$      ▷ Guidance correction step
6: **end for**
7: **return** $\boldsymbol{x}_N$

---

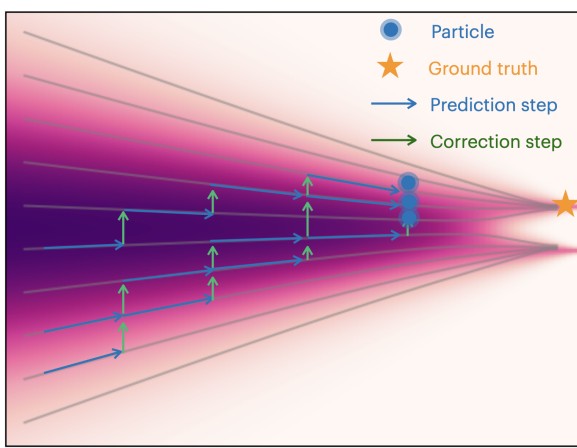

Figure 1: Illustration of the prediction-correction interpretation for guidance-based methods on a 1D Gaussian mixture example. From left to right, the probability flow ODE transforms $p_t(\boldsymbol{x}_t)$ from a Gaussian into a mixture of two Gaussians. The grey lines represent the trajectories of the probability flow. The prediction step corresponds to a numerical integration step along this flow. The correction step adjusts the particle towards the MAP estimator while keeping them close to the initial prediction point. In contrast to prior methods with independent particles, the proposed EnKG introduces interactions between particles to eliminate the need for gradient access.

**Prior prediction step**  This step is simply a numerical integration step of the unconditional probability flow ODE, i.e., by moving one step along the unconditional ODE trajectory:

$$\boldsymbol{x}_i' = \boldsymbol{x}_i - \dot{\sigma}(t_i)\sigma(t_i)s_\theta\left(\boldsymbol{x}_i, t_i\right)\left(t_{i+1} - t_i\right). \tag{5}$$

**Log-likelihood estimation step**  This step estimates the log-likelihood $\log p(\boldsymbol{y}|\boldsymbol{x}_t)$:

$$\log \hat{p}(\boldsymbol{y}|\boldsymbol{x}_i) \approx \log p(\boldsymbol{y}|\boldsymbol{x}_i).$$

**Guidance correction step**  This step solves the following optimization problem that formulates a compromise between maximizing the log-likelihood and being near $\boldsymbol{x}\prime_i$:

$$\boldsymbol{x}_{i+1} = \arg\min_{\boldsymbol{x}_{i+1}} \frac{\|\boldsymbol{x}_{i+1} - \boldsymbol{x}_i'\|_2^2}{2w_i} - \log \hat{p}(\boldsymbol{y}|\boldsymbol{x}_{i+1}), \tag{6}$$

where the larger guidance scale $w_i$ gives the solution point near the MAP estimator and smaller value leads to small movement towards the MAP estimator. Eq. (6) is essentially a proximal operator (Parikh et al., 2014) if $w_i$ is lower bounded by a positive number. This optimization problem is often non-convex in most practical scenarios. As a result, the optimization algorithm may converge to a local maximum rather than a global one.

To solve Eq. (6) efficiently, one can use a first-order Taylor approximation of $\log \hat{p}(\boldsymbol{y}|\boldsymbol{x}_{i+1})$ at $\boldsymbol{x}_i'$:

$$\log \hat{p}(\boldsymbol{y}|\boldsymbol{x}_{i+1}) = \log \hat{p}(\boldsymbol{y}|\boldsymbol{x}_i') + \nabla_{\boldsymbol{x}}^\top \log \hat{p}(\boldsymbol{y}|\boldsymbol{x}_i')\left(\boldsymbol{x}_{i+1} - \boldsymbol{x}_i'\right) + O\left(|\boldsymbol{x}_{i+1} - \boldsymbol{x}_i|^2\right). \tag{7}$$

---

**Algorithm 2** Our method: Ensemble Kalman Diffusion Guidance (EnKG).

---

**Require:** $G, \boldsymbol{y}, s_\theta, \text{solver } \phi, \{t_i\}_{i=1}^N, \{w_i\}_{i=1}^N, J$

1: **sample** $\boldsymbol{x}_0^{(j)} \sim \mathcal{N}(0, \sigma^2(t_0)\boldsymbol{I}), j = 1, \ldots, J$        ▷ Initialize particles

2: **for** $i \in \{0, \ldots, N-1\}$ **do**

3:      $\boldsymbol{x}_i'^{(j)} \leftarrow \boldsymbol{x}_i^{(j)} - \dot{\sigma}(t_i)\sigma(t_i)s_\theta\left(\boldsymbol{x}_i^{(j)}, t_i\right)(t_{i+1} - t_i)$        ▷ Prior prediction step

4:      $\hat{\boldsymbol{x}}_N^{(j)} \leftarrow \phi\left(\boldsymbol{x}_i^{(j)}, t_i\right), j = 1, \ldots, J$

5:      $g_i^{(j)} \leftarrow \frac{1}{J}\sum_{k=1}^J \left\langle G(\hat{\boldsymbol{x}}_N^{(k)}) - \bar{G}, \boldsymbol{y} - G(\hat{\boldsymbol{x}}_N^{(j)})\right\rangle_\Gamma \left(\boldsymbol{x}_i^{(k)} - \bar{\boldsymbol{x}}_i\right)$

6:      $\boldsymbol{x}_{i+1}^{(j)} \leftarrow \boldsymbol{x}_i'^{(j)} + w_i g_i^{(j)}, j = 1, \ldots, J$        ▷ Guidance correction step

7: **end for**

8: **return** $\{\boldsymbol{x}_N^{(j)}\}_{j=1}^J$

---

Substituting the approximation Eq. (7) into the correction step (6) gives:

$$\boldsymbol{x}_{i+1} \approx \arg\min_{\boldsymbol{x}_{i+1}} \frac{\|\boldsymbol{x}_{i+1} - \boldsymbol{x}_i'\|_2^2}{2w_i} - \log \hat{p}(\boldsymbol{y}|\boldsymbol{x}_i') - \nabla_{\boldsymbol{x}}^\top \log \hat{p}(\boldsymbol{y}|\boldsymbol{x}_i')(\boldsymbol{x}_{i+1} - \boldsymbol{x}_i') \tag{8}$$

$$= \boldsymbol{x}_i' + w_i \nabla_{\boldsymbol{x}} \log \hat{p}(\boldsymbol{y}|\boldsymbol{x}_i'), \tag{9}$$

which can recover the gradient step structure of most existing guidance-based methods (Chung et al., 2022a; Song et al., 2023b;a; Mardani et al., 2023).

**Putting it together.** Figure 1 depicts the Prediction-Correction interpretation in a 1D Gaussian mixture example, where guidance-based methods iteratively step towards the MAP estimator while staying close to the initial unconditional generation trajectory defined by the prediction step. Importantly, the PC framework allows more degrees of freedom in method design.

## 4.2 Our approach: Ensemble Kalman Diffusion Guidance

We now demonstrate how the correction step can be performed in a derivative-free manner using the idea of statistical linearization. Our overall approach is described in Algorithm 2.

**Likelihood estimation.** The likelihood term can be factorized as follows:

$$p(\boldsymbol{y}|\boldsymbol{x}_i) = \int p(\boldsymbol{y}|\boldsymbol{x}_N)p(\boldsymbol{x}_N|\boldsymbol{x}_i)\mathrm{d}\boldsymbol{x}_N = \mathbb{E}_{\boldsymbol{x}_N \sim p(\boldsymbol{x}_N|\boldsymbol{x}_i)}p(\boldsymbol{y}|\boldsymbol{x}_N), \tag{10}$$

which is intractable in general. We use the following Monte Carlo approximation:

$$p(\boldsymbol{y}|\boldsymbol{x}_i) = \mathbb{E}_{\boldsymbol{x}_N \sim p(\boldsymbol{x}_N|\boldsymbol{x}_i)}p(\boldsymbol{y}|\boldsymbol{x}_N) \approx p(\boldsymbol{y}|\hat{\boldsymbol{x}}_N), \tag{11}$$

where $\hat{\boldsymbol{x}}_N$ is obtained by running the PF ODE solver $\phi$ starting at $\boldsymbol{x}_i$. One attractive property of this estimate compared to popular ones based on $\mathbb{E}[\boldsymbol{x}_N|\boldsymbol{x}_i]$ and isotropic Gaussian approximations in previous works Chung et al. (2022a); Song et al. (2023a;b) is that our approximation stays on data manifold but the Gaussian approximations include additive noise that do not live on data manifold. This aspect is particularly important for scientific inverse problems based on partial differential equations (PDEs), where staying on the data manifold is important for reliably solving the forward model $p(\boldsymbol{y}|\boldsymbol{x})$. For instance, we observe that Gaussian approximations frequently violate the stability conditions of numerical PDE solvers, leading to meaningless estimates.

**Derivative-free correction step.** Consider an ensemble of particles $\{\boldsymbol{x}_i^{(j)}\}_{j=1}^J$. Let $\bar{\boldsymbol{x}}_i$ denote their empirical mean and $C_{xx}^{(i)}$ denote their empirical covariance matrix, at the $i$-th iteration:

$$\bar{\boldsymbol{x}}_i = \frac{1}{J}\sum_{j=1}^J \boldsymbol{x}_i^{(j)}, \quad C_{xx}^{(i)} = \frac{1}{J}\sum_{j=1}^J \left(\boldsymbol{x}_i^{(j)} - \bar{\boldsymbol{x}}_i\right)\left(\boldsymbol{x}_i^{(j)} - \bar{\boldsymbol{x}}_i\right)^\top.$$

Instead of the commonly used scalar weight $w_i$, we use a weighting matrix $w_i C_{xx}^{(i)}$ in Eq. (8):

$$\boldsymbol{x}_{i+1}^{(j)} \approx \arg\min_{\boldsymbol{x}_{i+1}} \frac{1}{2}\left(\boldsymbol{x}_{i+1} - \boldsymbol{x}_i'^{(j)}\right)^\top \left(w_i C_{xx}^{(i)}\right)^{-1}\left(\boldsymbol{x}_{i+1} - \boldsymbol{x}_i'^{(j)}\right) \tag{12}$$

$$- \nabla_{\boldsymbol{x}}^\top \log \hat{p}\left(\boldsymbol{y}|\boldsymbol{x}_i'^{(j)}\right)\left(\boldsymbol{x}_{i+1} - \boldsymbol{x}_i'^{(j)}\right) \tag{13}$$

$$= \boldsymbol{x}_i'^{(j)} + w_i C_{xx}^{(i)}\nabla_{\boldsymbol{x}}\log\hat{p}\left(\boldsymbol{y}|\boldsymbol{x}_i'^{(j)}\right). \tag{14}$$

Note that in practice, $C_{xx}^{(i)}$ can be singular when the number of particles is smaller than the particle dimension. In such cases, the matrix inverse in Eq. (12) is generalized to the sense of the Moore-Penrose inverse as $C_{xx}^{(i)}$ is always positive semi-definite. Eq. (14) effectively becomes a gradient projected onto the subspace spanned by the particles. At its current form, Eq. (14) still requires the gradient information. Next, we show how to approximate this gradient step without explicit derivative by Leveraging the idea of statistical linearization in the ensemble Kalman methods (Bergemann & Reich, 2010; Schillings & Stuart, 2017).

**Assumption 1.** *$G \circ \phi$ has bounded first and second order derivatives. Let $\psi$ denote $G \circ \phi$. There exist constants $M_1, M_2$ such that for all $\boldsymbol{u}, \boldsymbol{u}', \boldsymbol{v}, \boldsymbol{v}' \in \mathbb{R}^d$,*

$$\|\psi(\boldsymbol{u}) - \psi(\boldsymbol{u}')\| \le M_1\|\boldsymbol{u} - \boldsymbol{u}'\|, \boldsymbol{v}^\top H_\psi(\boldsymbol{v}')\boldsymbol{v} \le M_2\|\boldsymbol{v}\|^2.$$

*where $H_\psi$ denotes the Hessian matrix of $\psi$.*

**Assumption 2.** *The distance between ensemble particles is bounded. There exists a constant $M_3$ such that $\|\boldsymbol{x}_i^{(j)} - \bar{\boldsymbol{x}}_i\| < M_3, j = 1, \ldots, J$.*

**Assumption 3.** *The observation empirical covariance matrix does not degenerate to zero unless the covariance matrix collapses to zero. In other words, $tr\left(C_{yy}^{(i)}\right) = 0$ if and only if $C_{xx}^{(i)} = 0$, and*

$$C_{xx}^{(i)} \ne 0 \rightarrow tr\left(C_{yy}^{(i)}\right) > M_4, M_4 > 0,$$

*where*

$$C_{yy}^{(i)} = \frac{1}{J}\sum_{j=1}^J \left(\psi(\boldsymbol{x}_i^{(j)}) - \bar{\psi}_i\right)\left(\psi(\boldsymbol{x}_i^{(j)}) - \bar{\psi}_i\right)^\top, \bar{\psi}_i = \frac{1}{J}\sum_{j=1}^J \psi(\boldsymbol{x}_i^{(j)}). \tag{15}$$

**Remark** To verify that Assumption 3 holds in most cases of interest, we plot the traces $tr(C_{xx})$ and $tr(C_{yy})$ across three experiments of different inverse problems. As shown in Figure 5, while larger $tr(C_{xx})$ does not always indicate larger $tr(C_{yy})$ due to the ill-posedness, $tr(C_{yy})$ approaches zero only when $tr(C_{xx})$ also approaches zero, aligning with our assumption.

**Proposition 1.** *Under Assumption 1, 2 and 3, suppose the correction step is implemented as follows with $w_i = 1/\left(tr\left(C_{yy}^{(i)}\right)\right)$,*

$$\boldsymbol{x}_{i+1}^{(j)} = \boldsymbol{x}_i'^{(j)} + w_i C_{xy}^{(i)}\left(\boldsymbol{y} - \psi\left(\boldsymbol{x}_i'^{(j)}\right)\right) \tag{16}$$

$$= \boldsymbol{x}_i'^{(j)} + w_i\frac{1}{J}\sum_{k=1}^J \left\langle\psi\left(\boldsymbol{x}_i'^{(k)}\right) - \bar{G}, \boldsymbol{y} - \psi\left(\boldsymbol{x}_i'^{(j)}\right)\right\rangle_\Gamma \left(\boldsymbol{x}_i'^{(k)} - \bar{\boldsymbol{x}}_i\right), \tag{17}$$

*where*

$$C_{xy}^{(i)} = \frac{1}{J}\sum_{j=1}^J \left(\boldsymbol{x}_i'^{(j)} - \bar{\boldsymbol{x}}_i\right)\left(\psi\left(\boldsymbol{x}_i'^{(j)}\right) - \bar{\psi}_i\right)^\top.$$

*After sufficient iterations, we have the following approximation:*

$$C_{xy}^{(i)}\left(\boldsymbol{y} - \psi\left(\boldsymbol{x}_i'^{(j)}\right)\right) = \frac{1}{J}\sum_{k=1}^{J}\left\langle \psi\left(\boldsymbol{x}_i'^{(k)}\right) - \bar{G}, \boldsymbol{y} - \psi\left(\boldsymbol{x}_i'^{(j)}\right)\right\rangle_\Gamma \left(\boldsymbol{x}_i'^{(k)} - \bar{\boldsymbol{x}}_i\right) \tag{18}$$

$$\approx C_{xx}^{(i)}\nabla_{\boldsymbol{x}}\log\hat{p}\left(\boldsymbol{y}|\boldsymbol{x}_i'^{(j)}\right), \tag{19}$$

*where*

$$\bar{G} = \frac{1}{J}\sum_{j=1}^{J}G\left(\hat{\boldsymbol{x}}_N^{(j)}\right) = \frac{1}{J}\sum_{j=1}^{J}\psi(\boldsymbol{x}_i'^{(j)}).$$

The detailed derivation can be found in Appendix A.2. Proposition 1 shows that the ensemble update step defined in Eq. (17) effectively approximates the preconditioned gradient step defined in Eq. (12) without explicit derivative:

$$\boldsymbol{x}_{i+1}^{(j)} = \boldsymbol{x}_i'^{(j)} + w_i C_{xy}^{(i)}\left(\boldsymbol{y} - \psi\left(\boldsymbol{x}_i'^{(j)}\right)\right) \approx \boldsymbol{x}_i'^{(j)} + w_i C_{xx}^{(i)}\nabla_{\boldsymbol{x}}\log\hat{p}\left(\boldsymbol{y}|\boldsymbol{x}_i'^{(j)}\right). \tag{20}$$

Algorithm 2 puts it all together and provides a complete description of the proposed method. Implementation details are provided in Appendix A.4.

## 5 Experiments

We empirically study our EnKG method on the classic image restoration problems and two scientific inverse problems. We view the scientific inverse problems as the more interesting domains for evaluating our method, particularly the Navier-Stokes equation where it is impractical to accurately compute the gradient of the forward model.

**Baselines** We focus on comparing against methods that only use black-box access to the forward model. The first two baselines, Forward-GSG and Central-GSG (Algorithm 3), use numerical estimation methods instead of automatic differentiation to approximate the gradient of the log-likelihood, and plug it into a standard gradient-based method, Diffusion Posterior Sampling (DPS) (Chung et al., 2023b). Specifically, Forward-CSG uses a forward Gaussian smoothed gradient (Eq. 37), and Central-CSG uses a central Gaussian smoothed gradient (Eq. 38). More details are in Appendix A.3. The last two baselines are Stochastic Control Guidance (SCG) (Huang et al., 2024) and Diffusion Policy Gradient (DPG) (Tang et al., 2024), discussed in Sec. 2. For Navier-Stokes, we also add the conventional Ensemble Kalman Inversion (EKI) (Iglesias et al., 2013).

### 5.1 Image inverse problems

Tackling image inverse problems (e.g., deblurring) is common in the literature and serves as a reasonable reference domain for evaluation. We note that we consider a harder version of the problem where we do not use the gradient of the forward model. Moreover, most imaging problems use a linear forward model (except for phase retrieval), which is significantly simpler than the non-linear forward models more often used in scientific domains.

**Problem setting** We evaluate our algorithm on the standard image inpainting, superresolution, deblurring (Gaussian), and phase retrieval problems. For inpainting, the forward model is a box mask with randomized location. For superresolution, we employ bicubic downsampling (either $\times 2$ or $\times 4$) and for Gaussian deblurring, a blurring kernel of size $61 \times 61$ with standard deviation 3.0. Finally, phase retrieval takes the magnitude of the Fourier transform of the image as the observation. We use measurement noise $\sigma = 0.05$ in all experiments except for superresolution on $64 \times 64$ images, where we set $\sigma = 0.01$. The pre-trained diffusion model for FFHQ $64 \times 64$ is taken unmodified from Karras et al. (2022). The model for FFHQ $256 \times 256$ is taken from Chung et al. (2022a) and converted to the EDM framework (Karras et al., 2022) using their Variance-Preserving (VP) preconditioning.

Table 1: Quantitative evaluation on FFHQ 256x256 dataset. We report average metrics for image quality and consistency on four tasks. Measurement noise is $\sigma = 0.05$ unless otherwise stated.

| | Inpaint (box) | | | SR ($\times 4$) | | | Deblur (Gauss) | | | Phase retrieval | | |
|---|---|---|---|---|---|---|---|---|---|---|---|---|
| | PSNR↑ | SSIM↑ | LPIPS↓ | PSNR↑ | SSIM↑ | LPIPS↓ | PSNR↑ | SSIM↑ | LPIPS↓ | PSNR↑ | SSIM↑ | LPIPS↓ |
| Forward-GSG | 17.82 | 0.562 | 0.302 | 18.08 | 0.469 | 0.384 | 24.43 | 0.704 | 0.206 | 7.88 | 0.070 | 0.838 |
| Central-GSG | 18.76 | 0.720 | 0.229 | 26.55 | 0.740 | 0.169 | 25.39 | 0.775 | 0.180 | 10.10 | 0.353 | 0.691 |
| DPG | 20.89 | **0.752** | **0.184** | **28.12** | **0.831** | **0.126** | **26.42** | **0.798** | **0.143** | 15.47 | 0.486 | 0.495 |
| SCG | 4.71 | 0.302 | 0.763 | 4.71 | 0.302 | 0.760 | 4.69 | 0.300 | 0.759 | 4.623 | 0.294 | 0.764 |
| EnKG(Ours) | **21.70** | 0.727 | 0.286 | 27.17 | 0.773 | 0.237 | 26.13 | 0.723 | 0.224 | **20.06** | **0.584** | **0.393** |

**Evaluation metrics** We evaluate the sample quality of all methods using peak signal signal-to-noise-ratio (PSNR), structural similarity (SSIM) index (Wang et al., 2004), and learned perceptual image patch similarity (LPIPS) score (Zhang et al., 2018).

**Results** We show the quantitative results in Table 1(Appendix), and qualitative results in Figure 7 (Appendix). On the easier linear inverse problems (inpainting, superresolution, and deblur), EnKG comes in second to DPG. On the harder non-linear phase retrieval problem, EnKG is comfortably the best approach. This trend is consistent with our results in the scientific inverse problems, which are all non-linear.

## 5.2 Navier-Stokes equation

The Navier-Stokes problem is representative of the key class of scientific inverse problems (Iglesias et al., 2013) that our approach aims to tackle. The gradient of the forward model is impractical to reliably compute via auto-differentiation, as it requires differentiating through a PDE solver. Having effective derivative-free methods would be highly desirable here.

**Problem setting** We consider the 2-d Navier-Stokes equation for a viscous, incompressible fluid in vorticity form on a torus, where $\boldsymbol{u} \in C\left([0,T]; H^r_{\text{per}}((0,2\pi)^2; \mathbb{R}^2)\right)$ for any $r > 0$ is the velocity field, $\boldsymbol{w} = \nabla \times \boldsymbol{u}$ is the vorticity, $\boldsymbol{w}_0 \in L^2_{\text{per}}\left((0,2\pi)^2; \mathbb{R}\right)$ is the initial vorticity, $\nu \in \mathbb{R}_+$ is the viscosity coefficient, and $f \in L^2_{\text{per}}\left((0,2\pi)^2; \mathbb{R}\right)$ is the forcing function. The solution operator $\mathcal{G}$ is defined as the operator mapping the vorticity from the initial vorticity to the vorticity at time $T$. $\mathcal{G}: \boldsymbol{w}_0 \to \boldsymbol{w}_T$. Our experiments implement it as a pseudo-spectral solver (He & Sun, 2007).

$$\begin{aligned}
\partial_t \boldsymbol{w}(\boldsymbol{x},t) + \boldsymbol{u}(\boldsymbol{x},t) \cdot \nabla \boldsymbol{w}(\boldsymbol{x},t) &= \nu \Delta \boldsymbol{w}(\boldsymbol{x},t) + f(\boldsymbol{x}), & \boldsymbol{x} &\in (0,2\pi)^2, t \in (0,T] \\
\nabla \cdot \boldsymbol{u}(\boldsymbol{x},t) &= 0, & \boldsymbol{x} &\in (0,2\pi)^2, t \in [0,T] \\
\boldsymbol{w}(\boldsymbol{x},0) &= \boldsymbol{w}_0(\boldsymbol{x}), & \boldsymbol{x} &\in (0,2\pi)^2
\end{aligned} \tag{21}$$

The goal is to recover the initial vorticity field from a noisy sparse observation of the vorticity field at time $T = 1$. Eq. (21) does not admit a closed form solution and thus there is no closed form derivative available for the solution operator. Moreover, obtaining an accurate numerical derivative via automatic differentiation through the numerical solver is challenging due to the extensive computation graph that can span thousands of discrete time steps. We first solve the equation up to time $T = 5$ using initial conditions from a Gaussian random field, which is highly complicated due to the non-linearity of Navier-Stokes equation. We sample 20,000 vorticity fields to train our diffusion model. Then, we independently sample 10 random vorticity fields as the test set.

**Evaluation metrics** We report the relative $L^2$ error to evaluate the accuracy of the algorithm, which is given by $\frac{\|\hat{\boldsymbol{w}}_0 - \boldsymbol{w}_0^*\|_{L^2}}{\|\boldsymbol{w}_0^*\|_{L^2}}$ where $\hat{\boldsymbol{w}}_0$ is the predicted vorticity and $\boldsymbol{w}_0^*$ is the ground truth. To comprehensively analyze the computational requirements of inverse problem solvers, we use the following metrics: the total number of forward model evaluations (Total # Fwd); the number of sequential forward model evaluations (Seq. # Fwd), where each evaluation depends on the previous one.; the total number of diffusion model evaluations (Total # DM); the number of sequential diffusion model evaluations (Seq. # DM), which is

Table 2: Comparison on the Navier-Stokes inverse problem. Numbers in parentheses represent the sample standard deviation. Metrics to evaluate computation costs are defined in Sec. 5.2. ∗: one or two test cases are excluded from the results due to numerical instability. Runtime is reported on a single A100 GPU.

| | $\sigma_{\text{noise}} = 0$ | $\sigma_{\text{noise}} = 1.0$ | $\sigma_{\text{noise}} = 2.0$ | Computation cost | | | | |
| --- | --- | --- | --- | --- | --- | --- | --- | --- |
| | Relative L2 | Relative L2 | Relative L2 | Total # Fwd | Total # DM | Seq # Fwd | Seq # DM | Runtime |
| EKI | 0.577 (0.138) | 0.609 (0.119) | 0.673 (0.107) | 1024k | 0 | 0.50k | 0 | 121 mins |
| Forward-GSG | 1.687 (0.156) | 1.612 (0.173) | 1.454 (0.154) | 2049k | 1k | 1k | 1k | 105 mins |
| Central-GSG | 2.203* (0.314) | 2.117 (0.295) | 1.746 (0.191) | 2048k | 1k | 1k | 1k | 105 mins |
| DPG | 0.325 (0.188) | 0.408* (0.173) | 0.466 (0.171) | 4000k | 1k | 1k | 1k | 228 mins |
| SCG | 0.908 (0.600) | 0.928 (0.557) | 0.966 (0.546) | 384k | 384k | 0.75k | 1k | 119 mins |
| EnKG(Ours) | **0.120** (0.085) | **0.191** (0.057) | **0.294** (0.061) | 295k | 3342k | 0.14k | 1.3k | 124 mins |

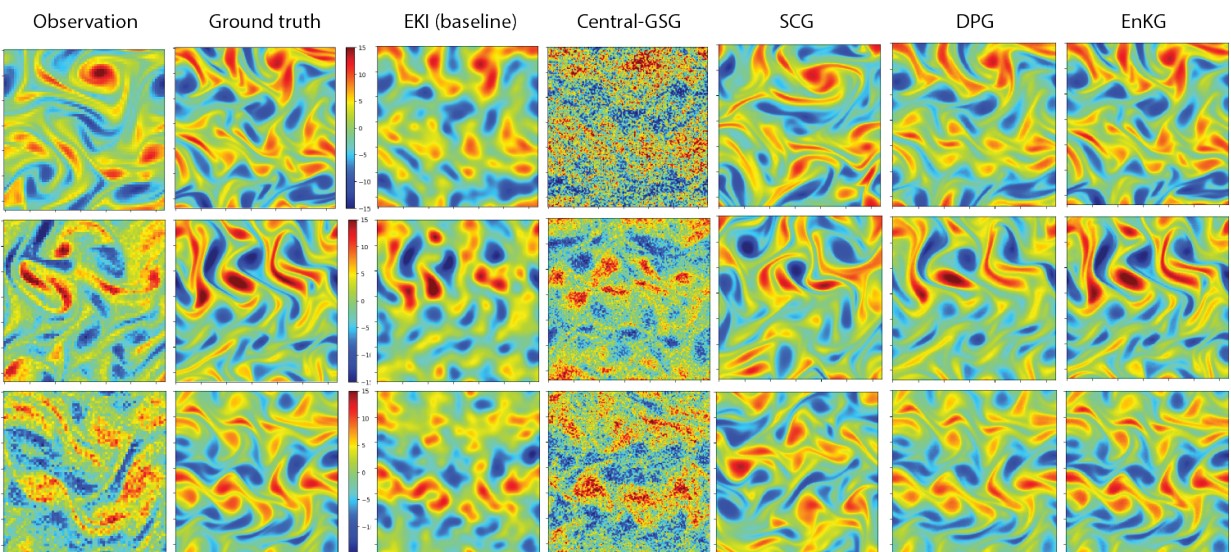

Figure 2: Visualization of results on Navier-Stokes inverse problem with different levels of observation noise. Each observation is subsampled by a factor of 2, representing a sparse measurement. Note that the results of Central-GSG are here for demonstration purpose because neither Central-GSG nor Forward-GSG is able to produce reasonable results.

analogous to Seq. # Fwd but focuses on diffusion model evaluation; the total number of diffusion model gradient evaluations (Total # DM grad); the number of sequential diffusion model gradient evaluations (Seq. # DM grad). These metrics are designed to reflect the primary computational demands: forward model queries and diffusion model queries. Sequential metrics are particularly important because they determine the minimum runtime achievable, independent of the available computational resources. By isolating sequential evaluations, we can better assess the parallelization potential of the algorithm, akin to the "critical path" concept in algorithm analysis from the computer science literature (Kohler, 1975).

**Results** In Table 2, we show the average relative $L^2$ error of the recovered ground truth at different noise levels of the observations. Our EnKG approach dramatically outperforms all other methods. Qualitatively, we see in Figure 2 that EnKG give solutions which qualitatively preserve important features of the flow, while some methods completely fail (i.e., overly noisy outputs).

On the computational aspect, the Navier-Stokes forward model (which requires a PDE solve) is extremely expensive, as shown in Figure 3(a). As such, the number of calls to the forward model dominates the computational cost. We see in Table 2 that our EnKG approach actually makes the fewest calls to the forward model (since it uses statistical linearization rather than trying to numerically approximate the gradient or value function), and thus EnKG is also the most computationally efficient approach, as seen in Figure 3(b). The traditional Ensemble Kalman Inversion (EKI) approach also employs statistical linearization, and so we

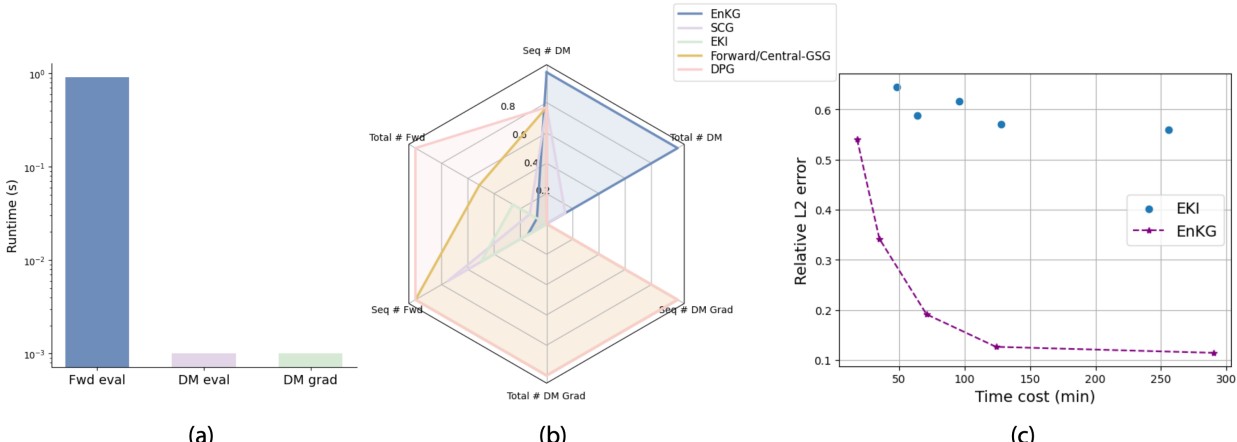

Figure 3: (a): runtime of single evaluation of the forward model, diffusion model, and diffusion model gradient (tested on a single A100). (b): comparison of computational characteristics of different algorithms on Navier-Stokes problem. Metrics are defined in the "Evaluation metrics" paragraph of Sec. 4.2. Each axis is normalized by dividing by the maximum over the algorithms. (c): compare EnKG with EKI on compute cost versus error.

do a detailed comparison in Figure 3(c), where we see that EnKG dominates EKI in the computational cost versus error trade-off curve.

**Ablation study** For practical insights into selecting hyperparameters for EnKG, we perform ablation studies on the ensemble size, guidance scale, and the number of diffusion model queries. The results are shown in Figure 6 and Figure 8. We observe a clear trend: increasing the ensemble size consistently leads to improved performance, as measured by the relative $L^2$ error. However, the gains become marginal beyond 2048 particles. Our experiments also reveal that a guidance scale of 2.0 yields the best average performance across our experiments. Increasing the guidance scale beyond this point introduces instability, resulting in higher relative $L^2$ errors. This instability is likely due to the fact that large guidance scale might lead to divergence as Lemma 1 only guarantees the convergence with small guidance scale. Furthermore, we vary the number of diffusion model queries using different number of ODE steps in the likelihood estimation $(\hat{\boldsymbol{x}}_N^{(j)})$. We also examine the effect of varying the number of diffusion model queries by changing the number of ODE steps used in estimating $\hat{\boldsymbol{x}}_N^{(j)}$. Performance improves with more diffusion model queries but exhibits diminishing returns beyond 500 steps.

## 5.3 Black-hole imaging inverse problem

The black-hole imaging problem is interesting due to its highly non-linear and ill-posed forward model (i.e., the sparse observations captured by telescopes on Earth). For evaluation purposes, we assume only black-box access to the forward model.

**Problem setting** The black hole interferometric imaging system aims to reconstruct image of black holes using a set of telescopes distributed on the Earth. Each pair of telescopes produces a measurement $V_{a,b}^t$ called *visibility*, where $(a, b)$ is a pair of telescopes and $t$ is the measuring time. To mitigate the effect of measurement noise caused by atmosphere turbulence and thermal noise, multiple visibilities can be grouped together to cancel out noise (Chael et al., 2018), producing noise-invariant measurements, termed closure phases $\boldsymbol{y}_{t,(a,b,c)}^{\text{cph}}$ and log closure amplitudes $\boldsymbol{y}_{t,(a,b,c,d)}^{\text{camp}}$. We specify the likelihood of these measurements similar to Sun & Bouman (2021):

$$\ell(\boldsymbol{y}|\boldsymbol{x}) = \sum_t \frac{\|\mathcal{A}_t^{\text{cph}}(\boldsymbol{x}) - \boldsymbol{y}_t^{\text{cph}}\|_2^2}{2\beta_{\text{cph}}^2} + \sum_t \frac{\|\mathcal{A}_t^{\text{camp}}(\boldsymbol{x}) - \boldsymbol{y}_t^{\text{camp}}\|_2^2}{2\beta_{\text{camp}}^2} + \rho \frac{\|\sum \boldsymbol{x}_i - \boldsymbol{y}^{\text{flux}}\|_2^2}{2}, \tag{22}$$

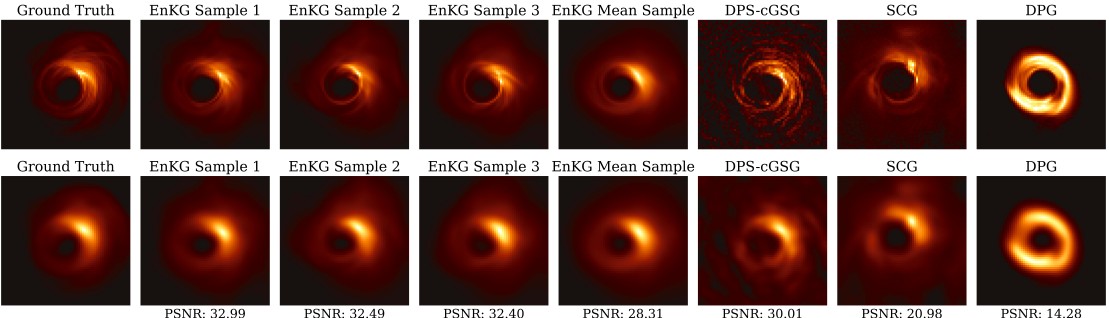

Figure 4: Visualization of generated samples on the black-hole imaging inverse problem. The first row shows the results on the original resolution, while the second row shows the blurred images in the intrinsic resolution of the imaging system.

Table 3: Quantitative evaluation of the reconstructed black-hole images.

|  | PSNR ↑ | Blurred PSNR ↑ | $\chi^2_{\text{cph}} \downarrow$ | $\chi^2_{\text{camp}} \downarrow$ |
|---|---|---|---|---|
| Central-GSG | 24.700 | 30.011 | 4.616 | 79.669 |
| SCG | 20.201 | 20.976 | **1.103** | **1.134** |
| DPG | 13.222 | 14.281 | 5.116 | 15.679 |
| EnKG (Ours) | **29.093** | **32.803** | 1.426 | 1.270 |

where $\mathcal{A}_t^{\text{cph}}$ and $\mathcal{A}_t^{\text{camp}}$ measures the closure phase and log closure amplitude of black hole images $\boldsymbol{x}$. $\beta_{\text{cph}}$ and $\beta_{\text{camp}}$ are known parameters from the telescope system. The first two terms are the sum of chi-square values for closure phases and log closure amplitudes, and the last term constrains the total flux of the black-hole image. We trained a diffusion model on the GRMHD dataset (Wong et al., 2022) with resolution $64 \times 64$ to generate black hole images from telescope measurements.

**Evaluation metrics** We report the chi-square errors of closure phases $\chi^2_{\text{cph}}$ and closure amplitudes $\chi^2_{\text{camp}}$ to measure how the generated samples fit the given measurement. We calculate the peak signal-to-noise ratio (PSNR) between reconstructed images and the ground truth. Moreover, since the black-hole imaging system provides only information for low spatial frequencies, following conventional evaluation methodology (EHTC, 2019), we blur images with a circular Gaussian filter and compute their PSNR on the intrinsic resolution of the imaging system.

**Results** Figure 4 shows the reconstructed images of the black-hole using our EnKG method and the baseline methods with black box access. EnKG is able to generate black hole images with visual features consistent with the ground truth. Table 3 shows the quantitative comparison. EnKG achieves relatively low measurement error (i.e., consistency with observations) and the best (blurred) PSNR compared with baseline methods (i.e., realistic images). SCG achieves slightly better data fitting metrics, but produces much noisier images than those by EnKG (Figure 4).

## 6 Conclusion and discussion

In this work, we propose EnKG, a fully derivative-free approach to solve general inverse problems that only permit black-box access. EnKG can accommodate different forward models without any re-training while maintaining the expressive ability of diffusion models to capture complex distribution. The experiments on various inverse problems arising from imaging and partial differential equations demonstrate the robustness and effectiveness of our methodology.

Despite its strengths, EnKG has certain limitations. First, as an optimization-based approach, it does not aim to recover the full posterior distribution and therefore cannot provide reliable uncertainty quantification—an important feature in some applications. Second, while EnKG reduces per-sample computational

cost compared to standard gradient-based methods, its total runtime is higher due to maintaining and updating an ensemble of interacting particles. Future work could explore adaptive strategies to dynamically adjust the number of particles based on the optimization landscape to improve efficiency. Additionally, integrating fast diffusion model sampling techniques (Zheng et al., 2023; Song et al., 2023c; Yin et al., 2024) may further reduce computational overhead.

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

Table 4: Table of notations.

| Notation | Description |
|---|---|
| $G$ | the forward model of the inverse problem |
| $\phi$ | Probability ODE solver |
| $\psi$ | Composition of $G$ and $\phi$ |
| $\boldsymbol{D}f$ | Jacobian matrix of function $f$ |
| $L^r_{\mathrm{per}}$ | Lebesgue space of periodic $r$-integrable functions |
| $H^r_{\mathrm{per}}$ | Sobolev space of $r$-times weakly differentiable periodic functions |
| $\Gamma$ | Covariance matrix of the Gaussian noise model |
| $\langle\cdot,\cdot\rangle_\Gamma$ | Weighted Euclidean inner product, $\langle\cdot,\cdot\rangle_\Gamma = \langle\cdot,\Gamma^{-1}\cdot\rangle$ |
| $\hat{\nabla}f$ | approximate gradient of $f$ |
| $\mu$ | Gaussian smoothing factor |
| $Q$ | number of gradient estimation queries |
| $w_i$ | log-likelihood gradient scale at step $i$ |
| $N$ | number of sampling steps |
| $E_{\mu,Q}(f(\boldsymbol{x}))$ | gradient estimator of $f(\boldsymbol{x})$ using smoothing factor $\mu$ and $Q$ queries |

# A Appendix / supplemental material

## A.1 Notation

## A.2 Proofs

**Lemma 1.** *Under Assumption 1, 2 and 3, suppose the correction step is implemented with $w_i = 1/\left(tr\left(C_{yy}^{(i)}\right)\right)$ as follows,*

$$\boldsymbol{x}_{i+1}^{(j)} = \boldsymbol{x}_i^{(j)} + w_i C_{xy}^{(i)}\left(\boldsymbol{y} - \psi\left(\boldsymbol{x}_i^{(j)}\right)\right),\tag{23}$$

*where $j \in \{1,\ldots,J\}$ and*

$$C_{xy}^{(i)} = \frac{1}{J}\sum_{j=1}^{J}\left(\boldsymbol{x}_i^{(j)} - \bar{\boldsymbol{x}}_i\right)\left(\psi\left(\boldsymbol{x}_i^{(j)}\right) - \bar{\psi}_i\right)^\top.$$

*Then $tr\left(C_{xx}^{(i)}\right)$ monotonically decreases to zero in the limit as $i$ goes to infinity.*

*Proof.* We first start from the ensemble update of the correction step given in Eq. (23) at iteration $i$ as follows

$$\boldsymbol{x}_{i+1}^{(j)} = \boldsymbol{x}_i^{(j)} + w_i C_{xy}^{(i)}\left(\boldsymbol{y} - \psi\left(\boldsymbol{x}_i^{(j)}\right)\right),\tag{24}$$

where $j \in \{1,\ldots,J\}$. The covariance matrix at the next iteration is given by

$$C_{xx}^{(i+1)} = \frac{1}{J}\sum_{j=1}^{J}(\boldsymbol{x}_{i+1}^{(j)} - \bar{\boldsymbol{x}}_{i+1})(\boldsymbol{x}_{i+1}^{(j)} - \bar{\boldsymbol{x}}_{i+1})^\top.\tag{25}$$

Plugging the update rule in Eq. (23) into Eq. (25), we have

$$
\begin{aligned}
C_{xx}^{(i+1)} =& \frac{1}{J} \sum_{j=1}^{J} \left[ (\boldsymbol{x}_i^{(j)} - \bar{\boldsymbol{x}}_i) + w_i C_{xy}^{(i)} \left( \bar{\psi}_i - \psi(\boldsymbol{x}_i^{(j)}) \right) \right] \left[ (\boldsymbol{x}_i^{(j)} - \bar{\boldsymbol{x}}_i) + w_i C_{xy}^{(i)} (\bar{\psi}_i - \psi(\boldsymbol{x}_i^{(j)})) \right]^{\top} \\
=& \frac{1}{J} \sum_{j=1}^{J} \left[ (\boldsymbol{x}_i^{(j)} - \bar{\boldsymbol{x}}_i)(\boldsymbol{x}_i^{(j)} - \bar{\boldsymbol{x}}_i)^{\top} + w_i^2 C_{xy}^{(i)} \left( \bar{\psi}_i - \psi(\boldsymbol{x}_i^{(j)}) \right) \left( \bar{\psi}_i - \psi(\boldsymbol{x}_i^{(j)}) \right)^{\top} C_{xy}^{(i)\top} \right] \\
& + \frac{1}{J} \sum_{j=1}^{J} \left[ w_i C_{xy}^{(i)} \left( \bar{\psi}_i - \psi(\boldsymbol{x}_i^{(j)}) \right) (\boldsymbol{x}_i^{(j)} - \bar{\boldsymbol{x}}_i)^{\top} + w_i (\boldsymbol{x}_i^{(j)} - \bar{\boldsymbol{x}}_i)(\bar{\psi}_i - \psi(\boldsymbol{x}_i^{(j)}))^{\top} C_{xy}^{(i)\top} \right]. \quad (26)
\end{aligned}
$$

We notice that

$$
\frac{1}{J} \sum_{j=1}^{J} w_i C_{xy}^{(i)} \left( \bar{\psi}_i - \psi(\boldsymbol{x}_i^{(j)}) \right) \left( \boldsymbol{x}_i^{(j)} - \bar{\boldsymbol{x}}_i \right)^{\top} = -w_i C_{xy}^{(i)} C_{xy}^{(i)\top}
$$

$$
\frac{1}{J} \sum_{j=1}^{J} w_i \left( \boldsymbol{x}_i^{(j)} - \bar{\boldsymbol{x}}_i \right) \left( \bar{\psi}_i - \psi(\boldsymbol{x}_i^{(j)}) \right)^{\top} C_{xy}^{(i)\top} = -w_i C_{xy}^{(i)} C_{xy}^{(i)\top}.
$$

Therefore, we can rewrite Eq. (26) as follows:

$$
C_{xx}^{(i+1)} = C_{xx}^{(i)} - 2w_i C_{xy}^{(i)} C_{xy}^{(i)\top} + w_i^2 C_{xy}^{(i)} C_{yy}^{(i)} C_{xy}^{(i)\top}.
$$

Further, by linearity of trace, we have

$$
tr\left( C_{xx}^{(i+1)} \right) = tr\left( C_{xx}^{(i)} \right) - 2w_i tr\left( C_{xy}^{(i)} C_{xy}^{(i)\top} \right) + w_i^2 tr\left( C_{xy}^{(i)} C_{yy}^{(i)} C_{xy}^{(i)\top} \right).
$$

By cyclic and submultiplicative properties, we have

$$
w_i^2 tr\left( C_{xy}^{(i)} C_{yy}^{(i)} C_{xy}^{(i)\top} \right) = w_i^2 tr\left( C_{yy}^{(i)} C_{xy}^{(i)\top} C_{xy}^{(i)} \right) \leq w_i^2 tr\left( C_{yy}^{(i)} \right) tr\left( C_{xy}^{(i)\top} C_{xy}^{(i)} \right).
$$

Since $w_i = 1/\left( tr\left( C_{yy}^{(i)} \right) \right)$, we have

$$
\begin{aligned}
tr\left( C_{xx}^{(i+1)} \right) &\leq tr\left( C_{xx}^{(i)} \right) - \frac{2}{tr\left( C_{yy}^{(i)} \right)} tr\left( C_{xy}^{(i)} C_{xy}^{(i)\top} \right) + \frac{1}{tr\left( C_{yy}^{(i)} \right)} tr\left( C_{xy}^{(i)\top} C_{xy}^{(i)} \right) \\
&= tr\left( C_{xx}^{(i)} \right) - \frac{1}{tr\left( C_{yy}^{(i)} \right)} tr\left( C_{xy}^{(i)} C_{xy}^{(i)\top} \right). \quad (27)
\end{aligned}
$$

By Assumption 1 and 2, we know that both $tr\left( C_{xx}^{(i)} \right)$ and $tr\left( C_{yy}^{(i)} \right)$ are upper bounded. By Assumption 3, $tr\left( C_{xy}^{(i)} C_{xy}^{(i)\top} \right)$ is lower bounded unless all the ensemble members collapse to a single point. Thus, there exists a $\alpha > 0$ such that $tr\left( C_{xy}^{(i)} C_{xy}^{(i)\top} \right) \geq \alpha \cdot tr\left( C_{xx}^{(i)} \right) tr\left( C_{yy}^{(i)} \right)$. Therefore,

$$
tr\left( C_{xx}^{(i+1)} \right) \leq tr\left( C_{xx}^{(i)} \right) - \frac{1}{tr\left( C_{yy}^{(i)} \right)} tr\left( C_{xy}^{(i)} C_{xy}^{(i)\top} \right) \leq (1 - \alpha) tr\left( C_{xx}^{(i)} \right).
$$

Note that from Eq. (27), we have $\alpha \leq 1$. Therefore, $tr\left( C_{xx}^{(i)} \right)$ monotonically decreases to zero. Additionally, we empirically check how quickly the average distance decays as we iterate in our experiments as shown in Figure 5. $\qquad\square$

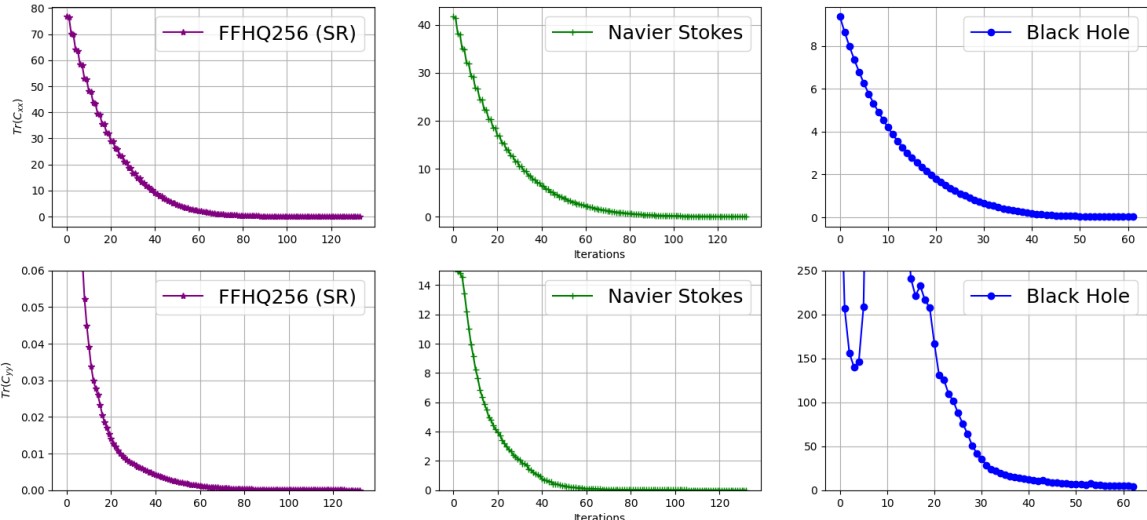

Figure 5: Empirical verification of Lemma 1 and Assumption 3. Top: the distance of ensemble members quickly decays over update steps. Bottom: while larger $tr(C_{xx})$ does not always indicate larger $tr(C_{yy})$, $tr(C_{yy})$ is close zero only when $tr(C_{xx})$ is close to zero.

**Proposition 1.** *Under Assumption 1, 2 and 3, suppose the correction step is implemented as follows with* $w_i = 1/\left(tr\left(C_{yy}^{(i)}\right)\right)$,

$$\boldsymbol{x}_{i+1}^{(j)} = \boldsymbol{x}_i'^{(j)} + w_i C_{xy}^{(i)}\left(\boldsymbol{y} - \psi\left(\boldsymbol{x}_i'^{(j)}\right)\right) \tag{28}$$

$$= \boldsymbol{x}_i'^{(j)} + w_i \frac{1}{J}\sum_{k=1}^{J}\left\langle \psi\left(\boldsymbol{x}_i'^{(k)}\right) - \bar{G}, \boldsymbol{y} - \psi\left(\boldsymbol{x}_i'^{(j)}\right)\right\rangle_\Gamma \left(\boldsymbol{x}_i'^{(j)} - \bar{\boldsymbol{x}}_i\right), \tag{29}$$

*where*

$$C_{xy}^{(i)} = \frac{1}{J}\sum_{j=1}^{J}\left(\boldsymbol{x}_i'^{(j)} - \bar{\boldsymbol{x}}_i\right)\left(\psi\left(\boldsymbol{x}_i'^{(j)}\right) - \bar{\psi}_i\right)^\top.$$

*After sufficient iterations, we have the following approximation:*

$$C_{xy}^{(i)}\left(\boldsymbol{y} - \psi\left(\boldsymbol{x}_i'^{(j)}\right)\right) = \frac{1}{J}\sum_{k=1}^{J}\left\langle \psi\left(\boldsymbol{x}_i'^{(k)}\right) - \bar{G}, \boldsymbol{y} - \psi\left(\boldsymbol{x}_i'^{(j)}\right)\right\rangle_\Gamma \left(\boldsymbol{x}_i'^{(j)} - \bar{\boldsymbol{x}}_i\right) \tag{30}$$

$$\approx C_{xx}^{(i)}\nabla_{\boldsymbol{x}}\log\hat{p}\left(\boldsymbol{y}|\boldsymbol{x}_i'^{(j)}\right). \tag{31}$$

*Proof.* Note that we can always normalize the problem so that $\Gamma$ is identity. Therefore, without loss of generality and for the ease of notation, we assume $\Gamma = \mathbf{I}$ throughout the whole proof. Given the inverse problem setting in Eq. 1 where the observation noise is Gaussian, we can rewrite the preconditioned gradient

w.r.t $\boldsymbol{x}_i'^{(j)}$ as

$$C_{xx}^{(i)} \nabla \log \hat{p} \left( \boldsymbol{y} | \boldsymbol{x}_i'^{(j)} \right) \tag{32}$$

$$= -\frac{1}{J} \sum_{k=1}^{J} \left( \boldsymbol{x}_i'^{(k)} - \bar{\boldsymbol{x}}_i \right) \left( \boldsymbol{x}_i'^{(k)} - \bar{\boldsymbol{x}}_i \right)^\top \nabla \frac{1}{2} \left\| \psi \left( \boldsymbol{x}_i'^{(j)} \right) - \boldsymbol{y} \right\|^2 \tag{33}$$

$$= -\frac{1}{J} \sum_{k=1}^{J} \left( \boldsymbol{x}_i'^{(k)} - \bar{\boldsymbol{x}}_i \right) \left( \boldsymbol{x}_i'^{(k)} - \bar{\boldsymbol{x}}_i \right)^\top \boldsymbol{D}^\top \psi \left( \boldsymbol{x}_i'^{(j)} \right) \left( \psi \left( \boldsymbol{x}_i'^{(j)} \right) - \boldsymbol{y} \right) \tag{34}$$

$$= -\frac{1}{J} \sum_{k=1}^{J} \left( \boldsymbol{x}_i'^{(k)} - \bar{\boldsymbol{x}}_i \right) \left( \boldsymbol{D}\psi \left( \boldsymbol{x}_i'^{(j)} \right) \boldsymbol{x}_i'^{(k)} - \boldsymbol{D}\psi \left( \boldsymbol{x}_i'^{(j)} \right) \bar{\boldsymbol{x}}_i \right)^\top \left( \psi \left( \boldsymbol{x}_i'^{(j)} \right) - \boldsymbol{y} \right) \tag{35}$$

$$= -\frac{1}{J^2} \sum_{k=1}^{J} \sum_{l=1}^{J} \left( \boldsymbol{x}_i'^{(k)} - \bar{\boldsymbol{x}}_i \right) \left( \boldsymbol{D}\psi \left( \boldsymbol{x}_i'^{(j)} \right) \left( \boldsymbol{x}_i'^{(k)} - \boldsymbol{x}_i^{(l)} \right) \right)^\top \left( \psi \left( \boldsymbol{x}_i'^{(j)} \right) - \boldsymbol{y} \right). \tag{36}$$

By definition, we have

$$tr \left( C_{xx}^{(i)} \right) = tr \left( \frac{1}{J} \sum_{j=1}^{J} \left( \boldsymbol{x}_i^{(j)} - \bar{\boldsymbol{x}}_i \right) \left( \boldsymbol{x}_i^{(j)} - \bar{\boldsymbol{x}}_i \right)^\top \right)$$

$$= \frac{1}{J} \sum_{j=1}^{J} tr \left( \left( \boldsymbol{x}_i^{(j)} - \bar{\boldsymbol{x}}_i \right)^\top \left( \boldsymbol{x}_i^{(j)} - \bar{\boldsymbol{x}}_i \right) \right)$$

$$= \frac{1}{J} \sum_{j=1}^{J} \| \boldsymbol{x}_i^{(j)} - \bar{\boldsymbol{x}}_i \|_2^2,$$

which represents the average distance between ensemble members. By Lemma 1, we know that $tr \left( C_{xx}^{(i)} \right)$ monotonically decreases to zero in the limit. Therefore, the ensemble members will get sufficiently close as we iterate. Therefore, we can apply first-order Taylor approximation to $\psi$ at $\boldsymbol{x}_i'^{(j)}$ under Assumption 1 and obtain

$$\psi \left( \boldsymbol{x}_i'^{(k)} \right) = \psi \left( \boldsymbol{x}_i'^{(j)} + \boldsymbol{x}_i'^{(k)} - \boldsymbol{x}_i'^{(j)} \right)$$

$$= \psi \left( \boldsymbol{x}_i'^{(j)} \right) + \boldsymbol{D}\psi \left( \boldsymbol{x}_i'^{(j)} \right) \left( \boldsymbol{x}_i'^{(k)} - \boldsymbol{x}_i'^{(j)} \right) + O \left( \| \boldsymbol{x}_i'^{(k)} - \boldsymbol{x}_i'^{(j)} \|_2^2 \right),$$

where $k \in \{1, \dots, J\}$. Therefore for any $k, l \in \{1, \dots, J\}$, by applying the approximation above at both $\boldsymbol{x}_i'^{(k)}$ and $\boldsymbol{x}_i'^{(l)}$, we have

$$\psi \left( \boldsymbol{x}_i'^{(k)} \right) - \psi \left( \boldsymbol{x}_i'^{(l)} \right) \approx \boldsymbol{D}\psi \left( \boldsymbol{x}_i'^{(j)} \right) \left( \boldsymbol{x}_i'^{(k)} - \boldsymbol{x}_i'^{(l)} \right)$$

We then plug it into Eq. 36

$$C_{xx}^{(i)} \nabla \log \hat{p} \left( \boldsymbol{y} | \boldsymbol{x}_i'^{(j)} \right)$$

$$\approx -\frac{1}{J^2} \sum_{k=1}^{J} \sum_{l=1}^{J} \left( \boldsymbol{x}_i'^{(k)} - \bar{\boldsymbol{x}}_i \right) \left( \psi \left( \boldsymbol{x}_i'^{(k)} \right) - \psi \left( \boldsymbol{x}_i^{(l)} \right) \right)^\top \left( \psi \left( \boldsymbol{x}_i'^{(j)} \right) - \boldsymbol{y} \right)$$

$$= -\frac{1}{J} \sum_{k=1}^{J} \left( \boldsymbol{x}_i'^{(k)} - \bar{\boldsymbol{x}}_i \right) \left( \psi \left( \boldsymbol{x}_i'^{(k)} \right) - \bar{\psi}_i \right)^\top \left( \psi \left( \boldsymbol{x}_i'^{(j)} \right) - \boldsymbol{y} \right)$$

$$= -\frac{1}{J} \sum_{k=1}^{J} \left\langle \psi \left( \boldsymbol{x}_i'^{(k)} \right) - \bar{\psi}_i, \psi \left( \boldsymbol{x}_i'^{(j)} \right) - \boldsymbol{y} \right\rangle \left( \boldsymbol{x}_i'^{(k)} - \bar{\boldsymbol{x}}_i \right)$$

$$= \frac{1}{J} \sum_{k=1}^{J} \left\langle G(\hat{\boldsymbol{x}}_N'^{(k)}) - \bar{G}, \boldsymbol{y} - G(\hat{\boldsymbol{x}}_N^{(k)}) \right\rangle \left( \boldsymbol{x}_i'^{(k)} - \bar{\boldsymbol{x}}_i \right),$$

---

**Algorithm 3** Central/Forward-GSG baseline with $\sigma(t) = t$ and $s(t) = 1$

---

**Require:** $G, \boldsymbol{y}, D_\theta, \{t_i\}_{i=1}^N, \{w_i\}_{i=1}^N, E_{\mu, Q}$

  1: **sample** $\boldsymbol{x}_0 \sim \mathcal{N}(0, t_0^2 \boldsymbol{I})$

  2: **for** $i \in \{0, \dots, N-1\}$ **do**

  3:      $\hat{\boldsymbol{x}}_0 \leftarrow D_\theta(\boldsymbol{x}_i, t_i)$

  4:      $\boldsymbol{x}_i' \leftarrow \boldsymbol{x}_i + \frac{\boldsymbol{x}_i - \hat{\boldsymbol{x}}_0}{t_i}(t_{i+1} - t_i)$                   ▷ Prior prediction step

  5:      $\hat{\nabla}_{\boldsymbol{x}_i} \log p(\boldsymbol{y}|\boldsymbol{x}_i) \leftarrow \nabla_{\boldsymbol{x}_i}(\hat{\boldsymbol{x}}_0^\top E_{\mu, Q}(\log p(\boldsymbol{y} \mid \hat{\boldsymbol{x}}_0)))$            ▷ Gradient estimation

  6:      $\boldsymbol{x}_{i+1} \leftarrow \boldsymbol{x}_i' + w_i \hat{\nabla}_{\boldsymbol{x}_t} \log p(\boldsymbol{y}|\boldsymbol{x}_i)$                ▷ Guidance correction step

  7: **end for**

  8: **return** $\boldsymbol{x}_N$

---

concluding the proof.                                                            □

### A.3   Zero-order gradient estimation baseline

We use the forward Gaussian smoothing and central Gaussian smoothing gradient estimation methods to establish a baseline to compare against. These methods approximate the gradient of a function using only function evaluations and can be expressed in the following (**Forward-GSG**) form :

$$\hat{\nabla} f(\boldsymbol{x}) = \sum_i^Q \frac{f(\boldsymbol{x} + \mu \boldsymbol{u}_i) - f(\boldsymbol{x})}{\mu} \tilde{\boldsymbol{u}}_i \tag{37}$$

And **Central-GSG**:

$$\hat{\nabla} f(\boldsymbol{x}) = \sum_i^Q \frac{f(\boldsymbol{x} + \mu \boldsymbol{u}_i) - f(\boldsymbol{x} - \mu \boldsymbol{u}_i)}{2\mu} \tilde{\boldsymbol{u}}_i \tag{38}$$

For Gaussian smoothing, $\boldsymbol{u}_i$ follows the standard normal distribution and $\tilde{\boldsymbol{u}}_i = \frac{1}{Q} \boldsymbol{u}_i$. The smoothing factor $\mu$ and number of queries $Q$ are both tunable hyperparameters.

Posterior sampling requires computation of the scores $\nabla_{\boldsymbol{x}_t} \log p(\boldsymbol{x}_t)$ and $\nabla_{\boldsymbol{x}_t} \log p(\boldsymbol{y} \mid \boldsymbol{x}_t)$; the former is learned by the pre-trained diffusion model, and the latter can be estimated by various approximation methods. In our baseline derivative-free inverse problem solver, we substitute the explicit automatic differentiation used in algorithms such as DPS with (37) and (38). We estimate this gradient by leveraging the fact that a probability flow ODE deterministically maps every $\boldsymbol{x}_t$ to $\boldsymbol{x}_0$; $\hat{\nabla}_{\hat{\boldsymbol{x}}_0} \log p(\boldsymbol{y} \mid \hat{\boldsymbol{x}}_0)$ is approximated with Gaussian smoothing, and a vector-Jacobian product (VJP) is used to then calculate $\hat{\nabla}_{\boldsymbol{x}_t} \log p(\boldsymbol{y} \mid \boldsymbol{x}_t)$. Our gradient estimate is defined as follows:

$$\hat{\nabla}_{\boldsymbol{x}_t} \log p(\boldsymbol{y} \mid \boldsymbol{x}_t) = \hat{\nabla}_{\boldsymbol{x}_t} \log p(\boldsymbol{y} \mid \hat{\boldsymbol{x}}_0) = \boldsymbol{D}_{\boldsymbol{x}_t}^\top \hat{\boldsymbol{x}}_0 \hat{\nabla}_{\hat{\boldsymbol{x}}_0} \log p(\boldsymbol{y} \mid \hat{\boldsymbol{x}}_0) \tag{39}$$

$\boldsymbol{D}_{\boldsymbol{x}_t}^\top$ is the transpose of the Jacobian matrix; (39) can be efficiently computed using automatic differentiation. Note that although automatic differentiation is used, differentiation through the forward model does not occur. Thus, this method is still applicable to non-differentiable inverse problems. Furthermore, we choose to perturb $\hat{\boldsymbol{x}}_0$ and use a VJP rather than directly perturb $\boldsymbol{x}_t$ so that we can avoid repeated forward passes through the pre-trained network, which is very expensive. Pseudocode for these algorithms is provided in Algorithm 3.

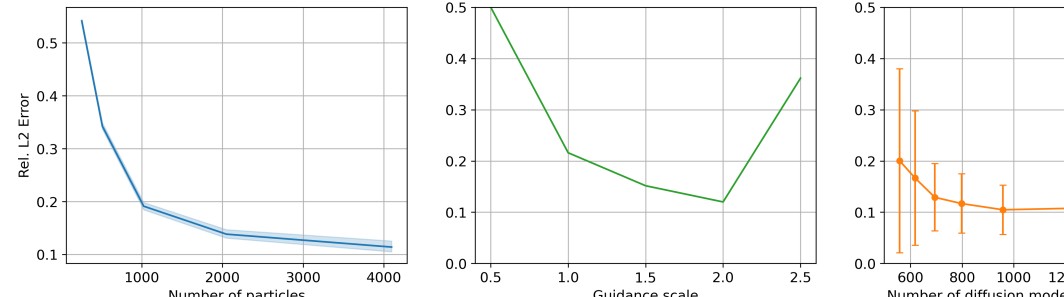

Figure 6: Ablation studies on the impact of ensemble size, guidance scale, and the number of diffusion model queries, conducted on the Navier-Stokes problem. Left: Relative $L^2$ error versus ensemble size; the shaded region indicates the range between the best and worst particle. Middle: Relative $L^2$ error versus the guidance scale. Right: Relative $L^2$ error versus the number of diffusion model queries per particle; error bars indicate one standard deviation.

Table 5: Qualitative evaluation on FFHQ 64x64 dataset. We report average metrics for image quality and samples consistency on four tasks. Measurement noise level $\sigma = 0.05$ is used if not otherwise stated.

| | Inpaint (box) | | | SR ($\times$2, $\sigma = 0.01$) | | | Deblur (Gauss) | | | Phase retrieval | | |
|---|---|---|---|---|---|---|---|---|---|---|---|---|
| | PSNR↑ | SSIM↑ | LPIPS↓ | PSNR↑ | SSIM↑ | LPIPS↓ | PSNR↑ | SSIM↑ | LPIPS↓ | PSNR↑ | SSIM↑ | LPIPS↓ |
| Forward-GSG | 19.62 | 0.612 | 0.189 | 25.25 | 0.836 | 0.093 | 20.27 | 0.606 | 0.170 | 10.307 | 0.170 | 0.493 |
| Central-GSG | 21.37 | 0.764 | 0.095 | 27.41 | 0.916 | 0.030 | 20.88 | 0.729 | 0.123 | 11.36 | 0.283 | 0.619 |
| DPG | 21.92 | 0.799 | 0.088 | 26.86 | 0.917 | **0.027** | 20.00 | 0.734 | 0.114 | 15.56 | 0.438 | 0.446 |
| SCG | 20.27 | 0.734 | 0.098 | 27.02 | 0.910 | 0.036 | 20.73 | **0.754** | **0.100** | 10.59 | 0.233 | 0.617 |
| EnKG(Ours) | **23.53** | **0.822** | **0.067** | **29.52** | **0.930** | 0.036 | **22.02** | 0.698 | 0.136 | **26.14** | **0.840** | **0.122** |

## A.4   EnKG Implementation Details

There are mainly two design choices in our algorithm 2 to be made. The first is the step size $w_i$ which controls the extent to which the correction step moves towards the MAP estimator. In the ensemble Kalman literature (Kovachki & Stuart, 2019), the following adaptive step size is widely used, and we adopt it for our experiments as well.

$$w_i^{-1} = \frac{1}{J^2} \sqrt{\sum_{k=1}^{J} \left\| G(\hat{\boldsymbol{x}}_N'^{(k)}) - \bar{G} \right\|^2 \left\| \boldsymbol{y} - G(\hat{\boldsymbol{x}}_N^{(j)}) \right\|^2} \tag{40}$$

Secondly, we find it useful to perform two correction steps in Eq. (6) when solving highly nonlinear and high-dimensional problems such as Navier Stokes. Therefore, we perform two correction steps at each iteration when running experiments on Navier Stokes.

## A.5   Baseline Details

## A.6   Additional results

We include more qualitative results for inverse problems on FFHQ 256x256 dataset in Figure 7.

|  | Observation | Ground truth | SCG | Forward-GSG | Central-GSG | DPG | EnKG |

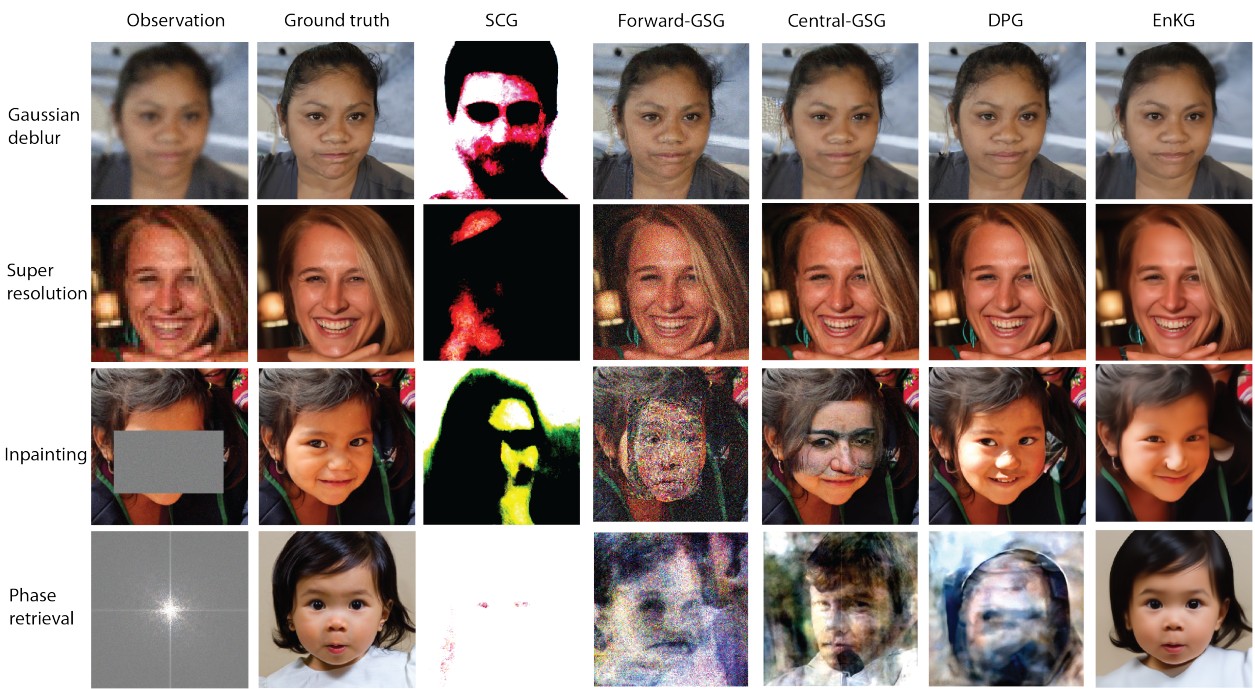

Figure 7: Qualitative results on FFHQ 256.

Table 6: Hyperparameter choices for Forward-GSG and Central-GSG ($64 \times 64$).

|  | Inpaint (box) | SR ($\times 2$, $\sigma = 0.01$) | Deblur (Gauss) | Phase retrieval |
|---|---|---|---|---|
| **Forward GSG** | | | | |
| $\mu$ | 0.001 | 0.001 | 0.001 | 0.001 |
| $Q$ | 10000 | 10000 | 10000 | 10000 |
| $w_i$ | 1.0 | 1.0 | 1.0 | 1.0 |
| $N$ | 1000 | 1000 | 1000 | 1000 |
| **Central GSG** | | | | |
| $\mu$ | 0.001 | 0.001 | 0.001 | 0.001 |
| $Q$ | 10000 | 10000 | 10000 | 10000 |
| $w_i$ | 1.0 | 1.0 | 1.0 | 1.0 |
| $N$ | 1000 | 1000 | 1000 | 1000 |

## A.7 Details of black hole imaging

The measurement of black hole imaging is defined as (Sun & Bouman, 2021)

$$\boldsymbol{y}_{t,(a,b,c)}^{\mathrm{cph}} = \angle(V_{a,b}^t V_{b,c}^t V_{a,c}^t) := \mathcal{A}_{t,(a,b,c)}^{\mathrm{cph}}(\mathbf{x}) \tag{41}$$

$$\boldsymbol{y}_{t,(a,b,c,d)}^{\mathrm{camp}} = \log\left(\frac{|V_{a,b}^t||V_{c,d}^t|}{|V_{a,c}^t||V_{b,d}^t|}\right) := \mathcal{A}_{t,(a,b,c,d)}^{\mathrm{camp}}(\boldsymbol{x}) \tag{42}$$

where $V_{a,b}$ is the visibility defined by

$$V_{a,b}^t(\boldsymbol{x}) = g_a^t g_b^t \exp(-i(\phi_a^t - \phi_b^t)) \cdot \tilde{\boldsymbol{I}}_{a,b}^t(\boldsymbol{x}) + \eta_{a,b}. \tag{43}$$

$g_a, g_b$ are telescope-based gain errors, $\phi_a^t, \phi_b^t$ are phase errors, and $\eta_{a,b}$ is baseline-based Gaussian noise. The measurements consist of $(M-1)(M-2)/2$ closure phases $\boldsymbol{y}^{\mathrm{cph}}$ and $M(M-3)/2$ log closure amplitudes $\boldsymbol{y}^{\mathrm{camp}}$ for an array of $M$ telescopes. Our experiments use $M = 9$ telescopes from Event Horizon Telescope.

Table 7: Hyperparameter choices for baselines Forward-GSG and Central-GSG ($256 \times 256$).

|  | Inpaint (box) | SR ($\times 4$, $\sigma = 0.05$) | Deblur (Gauss) | Phase retrieval |
|---|---|---|---|---|
| **Forward-GSG** |  |  |  |  |
| $\mu$ | 0.01 | 0.01 | 0.01 | 0.01 |
| $Q$ | 10000 | 10000 | 10000 | 10000 |
| $w_i$ | 1.0 | 1.0 | 3.0 | 0.7 |
| $N$ | 1000 | 1000 | 1000 | 1000 |
| **Central-GSG** |  |  |  |  |
| $\mu$ | 0.01 | 0.01 | 0.01 | 0.01 |
| $Q$ | 10000 | 10000 | 10000 | 10000 |
| $w_i$ | 1.0 | 1.0 | 3.0 | 0.7 |
| $N$ | 1000 | 1000 | 1000 | 1000 |

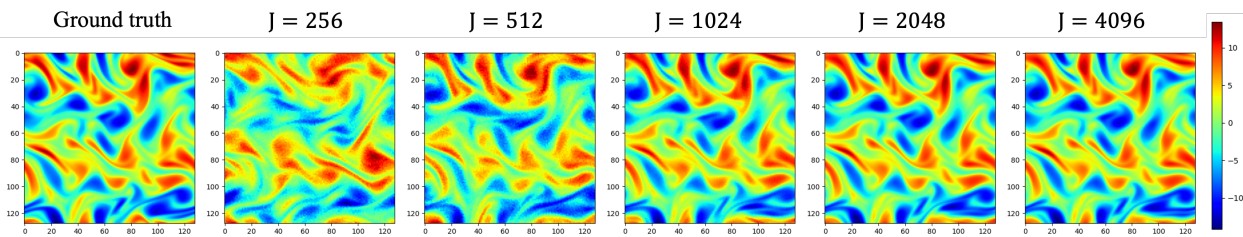

Figure 8: Vorticity field predicted by EnKG with different number of particles. From left to right, the result gets better as we increase the number of particles.

## A.8 Additional comparison

To provide a more comprehensive evaluation, we provide comparisons against several gradient-based methods across different tasks.

**Image restoration on FFHQ256** Table 8 presents comparisons with DPS (Chung et al., 2023b) and DiffPIR (Zhu et al., 2023) on four image restoration tasks: inpainting, super-resolution (x4), deblurring, and phase retrieval. We observe that EnKG achieves performance comparable to gradient-based methods, with no single approach emerging as a clear winner across all tasks. This demonstrates that EnKG offers competitive performance while maintaining its derivative-free property.

**Navier-Stokes equation** Table 9 reports comparison with DPS and PnP-DM (Wu et al., 2024) on the Navier-Stokes equation problem. We observe that EnKG clearly outperforms the gradient-based methods, while PnP-DM encounters numerical instability, resulting in either a crash or timeout. Since DPS and PnP-DM do not have such experiments in their paper, we perform a grid search for its guidance scale over range $[10^{-3}, 10^2]$ to find the best choice. For PnP-DM, we explore all hyperparameter combinations mentioned in their paper; however, all result in a numerical crash within the PDE solver. Although reducing the Langevin Monte Carlo learning rate improved stability, it led to infeasible runtimes (e.g., exceeding 100 hours). Consequently, we mark PnP-DM as "crashed/timeout" in Table 9. Additionally, in this problem, autograd encounters out-of-memory issues when the pseudospectral solver unrolls beyond approximately 6k steps on an A100-40GB GPU. This limitation suggests that gradient-based methods may not be applicable to more complex problems that require a large number of PDE solver iterations.

**Black hole imaging** As shown in Table 10 shows additional comparisons for the black hole imaging problem, including DPS and PnP-DM. Once again, EnKG delivers performance comparable to gradient-based methods. For DPS, we performed a grid search to optimize hyperparameters, while for PnP-DM, we used

the settings provided in their paper. These results further demonstrate the robustness and competitiveness of EnKG across diverse scientific inverse problems.

**Discussion**  As shown in the experiments above, gradient-based methods do not always perform better than derivative-free methods. We believe this is due to two reasons. First, gradient-based methods are more prone to getting stuck in local optima, especially in noisy, non-smooth settings. In contrast, many derivative-free methods, including EnKG, incorporate implicit smoothing, which can improve robustness in such cases. Second, gradient-based methods often rely on Gaussian approximations via Tweedie's formula, as taking gradients through the full unrolled reverse process is computationally infeasible. In contrast, EnKG gets a better likelihood approximation by simulating the reverse process without the need of backpropagation. More recent gradient-based works, such as Rout et al. (2024a); Chung et al. (2024), focus on solving linear inverse problems with text-to-image latent diffusion models. In contrast, EnKG does not rely on text-to-image latent diffusion model, which is generally unavailable in many inverse problem applications.

Table 8: Additional comparison with a few gradient-based methods on FFHQ 256x256 dataset. We report average metrics for image quality and consistency on four tasks. Measurement noise is $\sigma = 0.05$ unless otherwise stated.

| | Inpaint (box) | | | SR ($\times 4$) | | | Deblur (Gauss) | | | Phase retrieval | | |
|---|---|---|---|---|---|---|---|---|---|---|---|---|
| | PSNR↑ | SSIM↑ | LPIPS↓ | PSNR↑ | SSIM↑ | LPIPS↓ | PSNR↑ | SSIM↑ | LPIPS↓ | PSNR↑ | SSIM↑ | LPIPS↓ |
| **Gradient-based** | | | | | | | | | | | | |
| DPS | 21.77 | 0.767 | 0.213 | 24.90 | 0.710 | 0.265 | 25.46 | 0.708 | 0.212 | 14.14 | 0.401 | 0.486 |
| DiffPIR | 22.87 | 0.653 | 0.268 | 26.48 | 0.744 | 0.220 | 24.87 | 0.690 | 0.251 | 22.20 | 0.733 | 0.270 |
| **Black-box access** | | | | | | | | | | | | |
| EnKG(Ours) | 21.70 | 0.727 | 0.286 | 27.17 | 0.773 | 0.237 | 26.13 | 0.723 | 0.224 | 20.06 | 0.584 | 0.393 |

Table 9: Additional comparison of relative L2 error on the Navier-Stokes inverse problem. Numbers in parentheses represent the sample standard deviation.

| | $\sigma_{\text{noise}} = 0$ | $\sigma_{\text{noise}} = 1.0$ | $\sigma_{\text{noise}} = 2.0$ |
|---|---|---|---|
| **Gradient-based** | | | |
| DPS | 0.308 (0.214) | 0.349 (0.246) | 0.382 (0.228) |
| PnP-DM | Crashed or timeout | Crashed or timeout | Crashed or timeout |
| **Black-box access** | | | |
| EnKG(Ours) | 0.120 (0.085) | 0.191 (0.057) | 0.294 (0.061) |

Table 10: Additional comparison with a few gradient-based methods on the black-hole imaging problem.

| | PSNR ↑ | Blurred PSNR ↑ | $\chi^2_{\text{cph}} \downarrow$ | $\chi^2_{\text{camp}} \downarrow$ |
|---|---|---|---|---|
| DPS | 23.984 | 26.220 | 1.212 | 1.079 |
| PnP-DM | 28.211 | 32.499 | 1.120 | 1.224 |
| EnKG (Ours) | 29.093 | 32.803 | 1.426 | 1.270 |

### A.9  Robustness to the pretrained prior quality

In this section, we conduct a controlled experiment on Navier-Stokes equation problem to investigate the performance dependence on the quality of pre-trained diffusion models. Specifically, we trained a diffusion model prior using only 1/10 of the original training set and limited the training to 15k steps to simulate a lower-quality model. We evaluate the top two algorithms, EnKG and DPG, with the same hyperparameters used in the main experiments.

**Robust performance**  As shown in Table 11, we observe that while both algorithms experienced a performance drop due to the reduced quality of the diffusion model, the decline was relatively small compared to the significant reduction in training data. Notably, our EnKG demonstrated greater robustness, with a

smaller performance drop than the best baseline method, DPG. These results indicate that while EnKG benefits from high-quality diffusion models, it is not overly sensitive to their quality. It maintains strong performance even with reduced model capabilities.

Table 11: Relative L2 error of DPG and EnKG (ours) with different diffusion model quality.

|  | Original model trained with full data | New model trained with 1/10 data |
|---|---|---|
| DPG | 0.325 (0.188) | 0.394 (0.178) |
| EnKG (Ours) | 0.120 (0.085) | 0.169 (0.117) |

Table 12: Per-particle computational resource usage of EnKG across three inverse problems. We use 1024 particles for FFHQ and black hole imaging, and 2048 particles for Navier-Stokes. NFE-DM: number of diffusion model evaluations. NFE-Fwd: number of forward model evaluations. Runtime is averaged per particle.

|  | NFE-DM | NFE-Forward | Runtime (s) | Peak GPU memory (GB) |
|---|---|---|---|---|
| FFHQ 256×256 | 1632 | 144 | 11.6 | 23.9 |
| Navier-Stokes equation | 1632 | 144 | 3.6 | 7.1 |
| Black hole imaging | 771 | 60 | 1.1 | 1.4 |

