# OpenReview forum: "Ensemble Kalman Diffusion Guidance: A Derivative-free Method for Inverse Problems"
_TMLR — Accepted by TMLR_

### Review · Reviewer_huKD · 2025-02-23

**Summary Of Contributions:**

The paper proposed a new gradient-free method to solve the inverse problems. Specifically, it leveraged the advancement of both pre-trained diffusion model and the Ensemble Kalman methods. The key controbutions of this method is using black-box model without relying on gradient when they are inaccessable, as is in many real-world applications. The two-stage PC framework makes is feasible to avoid the gradient by using the forward model evaluations with the proposed ODE. The overall performance is good compared with other gradient-free alternatives.

**Audience:**

Yes

**Broader Impact Concerns:**

no concern.

**Claims And Evidence:**

Yes

**Requested Changes:**

1. More discussions of the relations to Ensemble Kalman methods.
2. More disussions with the other stronger gradient-based baselines as noted in the weakeness. Since gradient-based is not the focus, I'm more looking for discussions on the relations and explanations instead of additional experiments, now that it's mentioned in the paper alreday.

**Strengths And Weaknesses:**

Strengths:
1. The proposed method is novel and provide insights on applying agradient-free aproaches for the inverse problems.
2. The performace (both qualitative and quantitative) is good compared to other alternatives.
3. Experiment settings are comprehensive.

Weakness:
1. Needs more clarification for how you applied the motivations from the Ensemble Kalman methods since it's the core part of the paper, it will help the readers build the insight and relation on where the motivation come from to explain for "why this"?
2. Though the main focus is on gradient-free method, it good that the authors also comapre with more popular gradient-based method which helps building the bridge for the explanation, but more clarification is needed on the results, and more recent/compatitive baselines would be better for the comparison. For example STSL[1], P2L[2]. I would expect gradient-based models performs better but it doesn't seem to be the case, why?



[1] Rout, Litu, Yujia Chen, Abhishek Kumar, Constantine Caramanis, Sanjay Shakkottai, and Wen-Sheng Chu. "Beyond first-order tweedie: Solving inverse problems using latent diffusion." In Proceedings of the IEEE/CVF Conference on Computer Vision and Pattern Recognition, pp. 9472-9481. 2024.
[2] Chung, Hyungjin, Jong Chul Ye, Peyman Milanfar, and Mauricio Delbracio. "Prompt-tuning latent diffusion models for inverse problems." ICML 2025.

---

> ### Author Response · Authors · 2025-03-22
>
> **W1 & R1**
>
> We appreciate the reviewer's feedback on the need for a clearer explanation. Explanation of the core idea behind ensemble Kalman methods can be found in the global response. The explanation is also added to "Ensemble Kalman methods" paragraph of the "related work" section.
>
> **W2 & R2**
> We appreciate the reviewer’s constructive feedback and for suggesting additional references. We have included the following additional discussion in the updated appendix.
>
> **Gradient-based methods are not always superior**
> Even when the gradient is available and fast to compute, while having better computational efficiency, the gradient-based methods does not always have better performance.
> - Gradient-based methods are more prone to getting stuck in local optima, especially in noisy, non-smooth settings. In contrast, many derivative-free methods, including EnKG, incorporate implicit smoothing, which can improve robustness in such cases.
> - Gradient-based methods often rely on Gaussian approximations via Tweedie’s formula, as taking gradients through the full unrolled reverse process is computationally infeasible. In contrast, EnKG gets a better likelihood approximation by simulating the reverse process without the need of backpropagation.
>
> **Discussion about the provided references**
>
> Our paper differs from STSL and P2L:
> - STSL and P2L are designed for linear inverse problems as they require the adjoint matrix. In contrast, EnKG is applicable to a broader range of nonlinear problems, as demonstrated in our experiments. We also present it here for ease of review.
> - P2L and STSL are designed for latent diffusion models. Two key aspects of STSL, initialization and latent refinement, are tailored to latent diffusion models as they require the encoder. P2L requires a text-to-image diffusion model as it updates the text embedding during the inference. In contrast, EnKG does not rely on the text-to-image latent diffusion model, which is not available in many inverse problem applications.

---

### Review · Reviewer_uaeR · 2025-02-24

**Summary Of Contributions:**

The paper introduces Ensemble Kalman Diffusion Guidance (EnKG), a novel derivative-free approach for solving inverse problems using diffusion models. Unlike previous methods that rely on privileged information (e.g., derivatives or pseudo-inverses of the forward model), EnKG only requires black-box access to the forward model and a pre-trained diffusion model prior.

**Audience:**

Yes

**Claims And Evidence:**

Yes

**Requested Changes:**

The reviewers suggest the following improvements:
1.Investigate whether the number of particles in EnKG could be dynamically adjusted to improve efficiency.
2.Improve the writing and address the issues mentioned in Weakness 1.
3.Provide a detailed sensitivity analysis and ablation study that investigates how hyperparameters—particularly the ensemble size and guidance scaling factors—affect performance and computational cost.

**Strengths And Weaknesses:**

Strengths:
1.The work creatively combines ensemble Kalman filtering techniques with diffusion model guidance, addressing the limitation of requiring gradient or derivative information, especially in scenarios (e.g., PDE) where gradients are impractical to obtain.
2.The paper provides a clear derivation of the prediction–correction framework along with supporting theoretical results, which adds depth to the contribution.
3.The method is evaluated across a diverse set of inverse problems—from standard imaging tasks to challenging scientific problems—demonstrating its efficiency.

Weaknesses:
1.Writing needs improvement. (1). most readers are unfamiliar with the Ensemble Kalman and there should be a brief introduction to the core idea EnK (2) some symbols appear for the first time but are not explained, e.g., \hat{p}_t in Eq. 4
2.Although per-sample cost is reduced, maintaining an ensemble of particles can be computationally heavy and may not scale well in very high-dimensional settings.
3.The sensitivity of the method to key hyperparameters (such as ensemble size and guidance scaling) is not thoroughly explored, leaving questions about robustness and optimal settings.
4.Since the Prediction-correction framework is not new [1], you should clarify the significant differences between yours and [1].

[1] Score-based generative modeling through stochastic differential equations. ICLR 2021

---

> ### Author Response · Authors · 2025-03-22
>
> **W1 & R2**
> We thank the reviewer for the constructive feedback.
> - Explanation of the core idea behind ensemble Kalman methods: please see our explanation in the global response. The explanation is also added to "Ensemble Kalman methods" paragraph of the "related work" section.
> - We thank the reviewer for pointing out the missing definition.  We have now added the following explanation in the updated manuscript.
>     - The likelihood term, $p_t(y|x_t)=\int_{x_0} p(x_0|x_t)p(y|x_0)d x_0$, is computationally intractable as it requires integration over all possible $x_0$. Various tractable approximations have been proposed in the literature, which we denote as $\hat{p}_t(y|x_t)$.
>
>
> **W2**
> We appreciate the reviewer's concern regarding the scalability. We agree that EnKG has higher overall computation cost as it runs an ensemble of particles. However, due to its inherently parallel nature, it is as easy as data parallelism to scale up EnKG with the modern large-scale distributed computing infrastructure. Moreover, for scientific applications, one typically tolerate hours of computation to solve the inverse problems (e.g., in climate science).
>
>
> **W3 & R3**
> We appreciate the reviewer's concern. To address this, we have conducted ablation studies on the size of ensemble and guidance scale with the results presented in Figure 6 in the revised version.
>
> - Ensemble size: increasing the ensemble size consistently leads to improved performance, as measured by the relative L2 error. However, the gains become marginal beyond 2048 particles. The computational cost scales linearly with the ensemble size, which as noted before can be effectively parallized in practice.
> - Our experiments reveal that a guidance scale of 2.0 yields the best average performance across our experiments. Increasing the guidance scale beyond this point introduces instability, resulting in higher relative L2 errors. This instability is likely due to the fact that large guidance scale might lead to divergence as Lemma 1 only guarantees the convergence with small guidance scale. The guidance scale has no direct impact on computational cost.
>
>
>
>
> **W4**
> We appreciate the reviewer's feedback. We have added a paragraph about the difference from the existing PC methods in the "related work" section of the updated manuscript. We discuss the key differences from PC sampler in [1] below as a tailored response.
> - The PC sampler in [1] is only concerned with the diffusion model and aims to sample from the distribution learned by the diffusion model. Both the predictor and corrector aim to sample from the same target distribution by simulating different stochastic processes (reverse-time SDE and annealed Langevin dynamics, respectively).
> - Our Prediction-Correction framework is specific for finding solutions of inverse problems, which involves not only diffusion prior but also forward model and the observed measurement.
> - Our prediction step is a numerical integration over the probability flow ODE (PF-ODE) to integrate diffusion prior. The PC sampler in [1] could be incorporated in our prediction step for sampling from diffusion prior but the original PC sampler is not only inefficient but also unnecessarily complicated compared to the ODE solver.
> - Our correction step involves both the forward model $G$ and the observed measurement $y$. It optimizes the particles towards higher-likelihood region, which does not exist in [1].

---

> ### Author Response · Authors · 2025-03-22
> **Request 1**
>
> **R1**
> We appreciate the reviewer's suggestion regarding the adaptive adjustment of particle numbers in EnKG to improve efficiency. This is indeed an intriguing idea. To explore this idea, we performed experiments on Navier-Stokes equation problem with various strategies for varying the ensemble size during inference, under a fixed budget. Two key design decisions had to be addressed:
> 1. How to design the schedule of ensemble size?
>     - As a first attempt, We adopted a simple progressive increase scheme (512->1024->2048).
> 2. How to spawn new particles during inference? We explored several approaches
>     - SDEv1: new particles from performing a discrete step of the original reverse SDE in [1].
>     - SDEv2: new particles from performing a discrete step of the reverse SDE with smaller coefficient $\beta(t)=\frac{0.1}{\sigma(t)}$ in [2] (Eq. (6)).
>     - Gaussian approximation v1: new particles from sampling from the Gaussian approximation used in [3,4].
>     - Gaussian approrximation v2: new particles from Gaussian $N(\mathbb{E}[x_0|x_t], \sigma^2(t)I)$.
>
> | Scheme | Runtime |Relative L2 error |
> | -------- | -------- | -------- |
> | 1024 (constant) | 60 mins | 0.1913 |
> |512->1024->2048 SDEv1 | 60 mins | 0.2992|
> |512->1024->2048 SDEv2| 60 mins | 0.2800 |
> |512->1024->2048 Gaussian approximation v1 | 60 mins | 0.2939|
> |512->1024->2048 Gaussian approximation v2 | 60 mins | 0.2137|
>
> The results in the table show that the current constant-size scheme performs the best given the same runtime budget. While this is a preliminary investigation, it suggests that designing effective adaptive strategies is non-trivial and requires careful consideration of the aforementioned design choices.
>
> [2]: Karras, Tero, et al. "Elucidating the design space of diffusion-based generative models." Advances in neural information processing systems 35 (2022): 26565-26577.
>
> [3]: Song, Jiaming, et al. "Pseudoinverse-guided diffusion models for inverse problems." International Conference on Learning Representations. 2023.
>
> [4]: Song, Jiaming, et al. "Loss-guided diffusion models for plug-and-play controllable generation." International Conference on Machine Learning. PMLR, 2023.

---

> > ### Comment · Reviewer_uaeR · 2025-04-14
> > **2nd Comment**
> >
> > Thank you for your detailed response and the revisions made to the manuscript.
> >
> > We appreciate the added explanations and definitions improving clarity (W1/R2), the new ablation studies providing hyperparameter insights (Fig 6, W3/R3), the helpful clarification distinguishing your work from [1] (W4), and the preliminary experiments exploring adaptive ensembles (R1). We also note your points on scalability and parallelization (W2).
> >
> >
> > I now have two additional questions/suggestions:
> > 1.	In line 4, Algorithm2, $\hat{x}_N$ is obtained by running the Probability Flow ODE solver. Is this a complete ODE solution (hundreds of steps) or a one-step estimate (like DPS, Tweedie formula)? If the former, have you discussed reducing the number of ODE steps here (which can be different from N) to improve efficiency, which can refer to [2].
> > 2.	To further strengthen the paper's practical aspects, could you please include a table summarizing key computational resource usage for the main experiments? Specifically, reporting the Total NFE, Peak GPU Memory, and Total Computational Time (with respect to j) would be highly beneficial for the completeness of methodology.
> >
> > [2] Improving diffusion inverse problem solving with decoupled noise annealing. CVPR 2025.

---

> > > ### Author Response · Authors · 2025-04-17
> > > **Thanks for the valuable suggestion and question**
> > >
> > > **Number of ODE steps**
> > > Yes, we run PF-ODE solver to obtain $\hat{x}_N$, with the number of solver steps given by `(1 + (N - k) // 4)` at k-th iteration. We agree with the reviewer that reducing the number of ODE steps can improve efficiency, and we appreciate the pointer to [2]. In response, we conducted an additional ablation study to evaluate how the number of diffusion model queries affects performance. As shown in Figure 6 (right panel), we observe that EnKG can actually achieve the same performance with around half of the ODE steps,  resulting in approximately 25% total runtime savings. We sincerely thank the reviewer for this valuable suggestion.
> > >
> > > **Computational Resource Usage**
> > > As requested, we include a summary of per-particle computational resource usage in Table 12 of the updated manuscript. For ease of reference, we also reproduce the table below.
> > >
> > > Caption: Per-particle computational resource usage of EnKG across three inverse problems. We use 1024 particles for FFHQ and black hole imaging, and 2048 particles for Navier-Stokes. NFE-DM: number of diffusion model evaluations. NFE-Fwd: number of forward model evaluations. Runtime is averaged per particle.
> > >
> > > |  | NFE-DM | NFE-Forward |  Runtime per particle (s)   |  Peak GPU memory (GB)   |
> > > | -------- | ------ | ----------- | --- | --- |
> > > |  FFHQ 256x256   |    1632    |   144   |  11.6   | 23.9    |
> > > |  Navier-Stokes equation  |   1632  | 144   |  3.6   |  7.1   |
> > > | Black hole imaging     | 771   | 60    |   1.1  |  1.4   |
> > >
> > > *Note: The values reported in this table reflect the original configuration used in our main experiments. As shown in the additional ablation study (Figure 6, right), same performance can be achieved with approximately half the number of ODE steps. We are running experiments with this optimization and will update the numbers in the final version.*

---

### Review · Reviewer_STRX · 2025-03-09

**Summary Of Contributions:**

This paper introduces Ensemble Kalman Diffusion Guidance (EnKG), a novel derivative-free optimization method for solving inverse problems using pre-trained diffusion models. Unlike traditional methods, EnKG requires only black-box evaluations of the forward model, eliminating the need for explicit gradient computations or pseudo-inverses. It estimates guidance terms through statistical linearization from ensemble Kalman methods, approximating gradients using the empirical covariance matrix of ensemble particles along diffusion ODE trajectories. The effectiveness of EnKG is demonstrated across three types of inverse problems—computational imaging, fluid dynamics (Navier-Stokes equation), and black-hole imaging—highlighting its flexibility and robustness.

**Audience:**

Yes

**Claims And Evidence:**

Yes

**Requested Changes:**

I appreciate the paper's contributions. However, I believe it could be further improved by addressing the following.

 -  What distinguishes the predictor-corrector scheme presented in this paper from those described in existing works? Specifically, can you provide a comparative analysis and elucidate any connections to the following papers:
1)  Song, Yang, et al. "Score-Based Generative Modeling through Stochastic Differential Equations." ICLR 2021.
2) Lezama, Jose, et al. "Discrete predictor-corrector diffusion models for image synthesis." ECCV 2022.
3) Zhao, Wenliang, et al. "Unipc: A unified predictor-corrector framework for fast sampling of diffusion models." NeurIPS 2023.
4) Zhao, Wenliang, et al. "DC-Solver: Improving Predictor-Corrector Diffusion Sampler via Dynamic Compensation." ECCV 2024.
5) Bradley, Arwen, and Preetum Nakkiran. "Classifier-Free Guidance is a Predictor-Corrector." NeurIPSW 2024.

-  I question the plausibility of Assumption 3, particularly in low-dimensional and ill-posed operator scenarios. While it may hold true in the high-dimensional problems explored in this paper, I request plots illustrating the trace or singular values of the covariance matrices for x and y.

**Strengths And Weaknesses:**

### Strengths
- This paper introduces ensemble Kalman methods for diffusion-based inverse problems, a novel concept in this area.
- This paper proposes a unique guidance term formulation that employs statistical linearization, effectively replacing the derivative of the measurement operator with the covariance of evaluations produced by the measurement operator. This derivative-free approach is broadly applicable across various scenarios.
- The effectiveness of the proposed method is demonstrated in three key scenarios: (1) when the measurement model is known and differentiable, (2) when it is known but impractical to differentiate, such as in PDE-based models, and (3) when the measurement model is a black box, with only observational data available.

### Weaknesses
- The proposed method proves ineffective in situations where derivative computation is efficient while point evaluation is computationally costly.

---

> ### Author Response · Authors · 2025-03-22
>
> **W1**:
> Indeed, that is true.  However, the goal of this paper (as stated in the abstract and introduction) is to study the opposite scenario where derivative computation is intractable or impossible.  This setting applies to many real-world applications that involve forward models with complex, black-box structures, where derivative computation is either intractable or significantly more expensive than forward evaluation. Examples include simulations with complex physics or measurement filtering process. This is the setting where derivative-free methods like EnKG become valuable.
>
> **R1**:
> We appreciate the reviewer's suggestion and references. Please see the "Differences from existing PC methods" section in our global response. We have also added a paragraph about the difference from the existing PC methods in the "related work" section of the updated manuscript.
>
> **R2**:
> We thank the reviewer for the constructive advice for additional empirical validation. We have included plots of the trace of $C_{xx}$ and $C_{yy}$ at different time steps across all three inverse problems in Figure 5 (Page 17).
> - As shown in Figure 5, while larger $tr(C_{xx})$ does not always indicate larger $tr(C_{yy})$ due to the ill-posedness, $tr(C_{yy})$ approaches zero only when $tr(C_{xx})$ also approaches zero, aligning with our assumption.
> - We acknowledge that the ill-posedness of the problem allows for multiple possible particles to have the same measurement $y$. However, we argue that, for many applications, it is unlikely that  *all* the different particles in a sufficiently large ensemble (e.g. hundreds) have the same measurement $y$ unless they themselves also converge to the same point.

---

### Author Response · Authors · 2025-03-22
**Global response**

We sincerely thank the reviewers for their constructive feedback, insightful questions, and recognition of our contributions. Below, we summarize the key revisions made in response to the reviews. All changes are highlighted in cyan in the updated manuscript for easy reference.
- Added explantion of the idea behind ensemble Kalman methods to "related work" section (Section 3, page 3). (uaeR, huKD)
- Added discussion of the existing PC methods to "related work" section (Section 3, page 4). (STRX, uaeR)
- Added a remark after Assumption 3. (STRX)
- Figure 5: added the plots of $tr(C_{yy})$ to empirically validate Assumption 3. (STRX)
- Figure 6: added ablation studies on the ensemble size and guidance scale. (uaeR)
- Added a paragraph to discuss ablation studies (page 10). (uaeR)
- Appendix A.8: add a paragraph to discuss gradient-based methods. (huKD)

---

### Decision · Action_Editor_kyzf · 2025-05-27

**Recommendation:** Accept as is

**Comment:**

See above.

**Audience:**

Yes

**Claims And Evidence:**

In this paper, the authors propose a method leveraging pretrained diffusion model prior and (only) forward model evaluation in order to solve inverse problems. Their method called Ensemble Kalman Diffusion Guidance introduced a particle algorithm in order to compute the guidance. At a given time step they follow the ODE of the diffusion model and then compute a quantity based on the similarity with the observation which results in a non-scalar weight. The method is original and tested on several examples in image, fluid mechanics and black hole imaging.

All reviewers have recommended the acceptance for this work and the authors have answered most of the questions during the rebuttal period.